# Boosting Graph Pooling with Persistent Homology

**Chaolong Ying, Xinjian Zhao, Tianshu Yu***
School of Data Science, The Chinese University of Hong Kong, Shenzhen
{chaolongying,xinjianzhao1}@link.cuhk.edu.cn, yutianshu@cuhk.edu.cn

## Abstract

Recently, there has been an emerging trend to integrate persistent homology (PH) into graph neural networks (GNNs) to enrich expressive power. However, naively plugging PH features into GNN layers always results in marginal improvement with low interpretability. In this paper, we investigate a novel mechanism for injecting global topological invariance into pooling layers using PH, motivated by the observation that filtration operation in PH naturally aligns graph pooling in a cut-off manner. In this fashion, message passing in the coarsened graph acts along persistent pooled topology, leading to improved performance. Experimentally, we apply our mechanism to a collection of graph pooling methods and observe consistent and substantial performance gain over several popular datasets, demonstrating its wide applicability and flexibility.

## 1 Introduction

Persistent homology (PH) is a powerful tool in the field of topological data analysis, which is capable of evaluating stable topological invariant properties from unstructured data in a multi-resolution fashion [8]. Concretely, PH derives an increasing sequence of simplicial complex subsets by applying a filtration function (see Fig. 1(a)). According to the fact that PH is at least as expressive as Weisfeiler-Lehman (WL) hierarchy [19], there recently emerged a series of works seeking to merge PH into graph neural networks (GNNs), delivering competitive performance on specific tasks [52, 19]. Standard schemes of existing works achieve this by employing pre-calculated topological features [52] or placing learnable filtration functions in the neural architectures [16, 19]. Such integration of PH features is claimed to enable GNNs to emphasize persistent topological sub-structures. However, it is still unclear to what extent the feature-level integration of PH is appropriate and how to empower GNNs with PH other than utilizing features.

Graph pooling (GP) in parallel plays an important role in a series of graph learning methods [14], which hierarchically aggregates an upper-level graph into a more compact lower-level graph. Typically, GP relies on calculating an assignment matrix taking into account local structural properties such as community [34] and cuts [2]. Though the pooling paradigm in convolutional neural networks (CNNs) is quite successful [27], some researchers raise concerns about its effectiveness and applicability in graphs. For example, [30] challenges the local-preserving usage of GP by demonstrating that random pooling even leads to similar performance. Till now, it remains opaque what property should be preserved for pooled topology to better facilitate the downstream tasks.

From Fig. 1(a), it is readily observed that PH and GP both seek to coarsen/sparsify a given graph in a hierarchical fashion: while PH gradually derives persistent sub-topology (substructures such as cycles) by adjusting the filtering parameter, GP obtains a sub-graph by performing a more aggressive cut-off. In a sense of understanding a graph through a hierarchical lens, PH and GP turn out to align with each other well.

---

*Corresponding author

38th Conference on Neural Information Processing Systems (NeurIPS 2024).

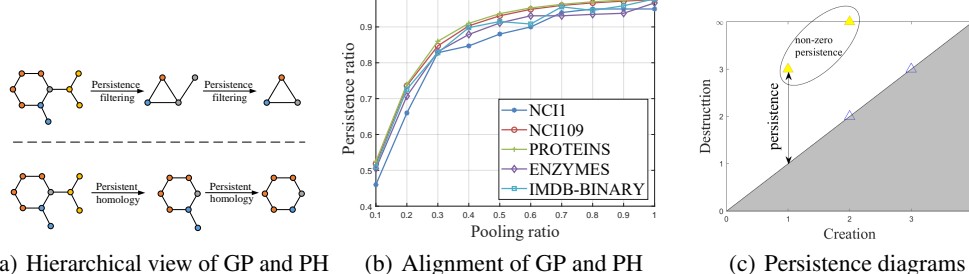

| (a) Hierarchical view of GP and PH | (b) Alignment of GP and PH | (c) Persistence diagrams |

Figure 1: Illustration of Graph Pooling (GP) and Persistent Homology (PH). (a) GP and PH share a similar hierarchical fashion by coarsening a graph. (b) As a motivating experiment, we gradually change pooling ratio and count how persistence ratio (ratio of non-zero persistence) changes with it. (c) Illustration of persistence diagrams.

Driven by this observation, in this paper, we investigate the mechanism of aligning PH and GP so as to mutually reinforce each other. To this end, we conduct experiments by running a pioneer GP method DiffPool [50] to conduct graph classification on several datasets and at the same time use the technique in [16] to compute PH information. We manually change the pooling ratio and see what proportion of meaningful topological information (characterized by the ratio of non-zero persistence) is naturally preserved at the final training stage. Surprisingly, the correspondence is quite stable regardless of different datasets (see Fig. 1(b)), which implies the monotone trend between the pooling ratio and non-zero persistence is commonly shared by a large range of graph data. As a consequence, we develop a natural way to integrate PH and GP in both feature and topology levels. Concretely, in addition to concatenating vectorized PH diagram as supplementary features, we further enforce the coarsened graph to preserve topological information as much as possible with a specially designed PH-inspired loss function. Hence we term our method Topology-Invariant Pooling (TIP). TIP can be flexibly injected into a variety of existing GP methods, and demonstrates a consistent ability to provide substantial improvement over them. We summarize our contributions as follows:

- We for the first time investigate the way of aligning PH with GP, by investigating the monotone relationship in between.

- We further design an effective mechanism to inject PH information into GP at both feature and topology levels, with a novel topology-preserving loss function.

- Our mechanism can be flexibly integrated with a variety of GP methods, achieving consistent and substantial improvement over multiple datasets.

## 2 Related work

**Graph pooling.** Graph pooling has been used in various applications, which can reduce the graph size while preserving its structural information. Early methods are based on clustering to coarsen graphs, such as the greedy clustering method Graclus [7], non-negative matrix factorization of the adjacency matrix [1], and spectral clustering [29]. Recently, learnable graph pooling methods have gained popularity, which learn to select important nodes in an end-to-end manner. DiffPool [50] follows a hierarchical learning structure by utilizing GNNs to learn clusters and gradually aggregate nodes into a coarser graph. MinCutPool [2] optimizes a normalized cut objective to partition graphs into clusters. DMoNPool [34] optimizes the modularity of graphs to ensure high-quality clusters. SEP [46] generates clusters in different hierarchies simultaneously without compromising local structures. These methods are classified as dense pooling due to the space complexity they incur. Despite their effectiveness, dense pooling methods have been criticized for high memory cost and complexity [5]. Therefore, various sparse pooling methods have been proposed, such as Top-K [11], ASAPool [38], and SAGPool [28]. These methods coarsen graphs by selecting a subset of nodes based on a ranking score. As they drop some nodes in the pooling process, these methods are criticized for their limited capacity to retain essential information, with potential effects on the expressiveness of preceding GNN layers [3].

**Persistent homology in GNNs.** PH is a technique to calculate topological features of structured data, and many approaches have been proposed to use PH in graph machine learning due to the high expressiveness of topological features on graphs [18]. Since non-isomorphic graphs may exhibit different topological features, the combination of PH and the Weisfeiler-Lehman (WL) algorithm leads to stronger expressive power [40]. This encourages further exploration on equipping GNNs with topological features. [52] propose that message passing in GNNs can be effectively reweighted using topological features. [16] and [19] provide theoretical and practical insights that filtrations in PH can be purely learnable, enabling flexible usage of topological features in GNNs. However, existing methods tend to view PH merely as a tool for providing supplementary information to GNNs, resulting in unsatisfactory improvements and limited interpretability.

## 3 Background

We briefly review the background of this topic in this section, as well as elaborate on the notations.

Let $\mathcal{G} = (V, E)$ be an undirected graph with $n$ nodes and $m$ edges, where $V$ and $E$ are the node and the edge sets, respectively. Nodes in attributed graphs are associated with features, and we denote by $V = \{(v, \mathbf{x}_v)\}_{v \in 1:n}$ the set of nodes $v$ with $d$ dimensional attribute $\mathbf{x}_v$. It is also practical to represent the graph with an adjacency matrix $\mathbf{A} \in \{0, 1\}^{n \times n}$ and the node feature matrix $\mathbf{X} \in \mathbb{R}^{n \times d}$.

**Graph Neural Networks.** We focus on the general message-passing GNN framework that updates node representations by iteratively aggregating information from neighbors [12]. Concretely, the $k$-th layer of such GNNs can be expressed as:

$$\mathbf{X}^{(k)} = \mathrm{M}\left(\mathbf{A}, \mathbf{X}^{(k-1)}; \theta^{(k)}\right),\tag{1}$$

where $\theta^{(k)}$ is the trainable parameter, and M is the message propagation function. Numbers of M have been proposed in previous research [25, 15]. A complete GNN is typically instantiated by stacking multiple layers of Eq. 1. Hereafter we denote by $\mathrm{GNN}(\cdot)$ an arbitrary such multi-layer GNN for brevity.

**Dense Graph Pooling.** GP in GNNs is a special layer designated to produce a coarsened or sparsified sub-graph. Formally, GP can be formulated as $\mathcal{G} \mapsto \mathcal{G}_P = (V_P, E_P)$ such that the number of nodes $|V_P| \leq n$. GP layers can be placed into GNNs in a hierarchical fashion to persistently coarsen the graph. Typical GP approaches [50, 2, 34] rely on learning a soft cluster assignment matrix $\mathbf{S}^{(l)} \in \mathbb{R}^{n_{l-1} \times n_l}$:

$$\mathbf{S}^{(l)} = \mathrm{softmax}\left(\mathrm{GNN}^{(l)}\left(\mathbf{A}^{(l-1)}, \mathbf{X}^{(l-1)}\right)\right).\tag{2}$$

Subsequently, the coarsened adjacency matrix at the $l$-th pooling layer is calculated as

$$\mathbf{A}^{(l)} = \mathbf{S}^{(l)\top}\mathbf{A}^{(l-1)}\mathbf{S}^{(l)},\tag{3}$$

and the corresponding node representations are calculated as

$$\mathbf{X}^{(l)} = \mathbf{S}^{(l)\top}\mathrm{GNN}^{(l)}\left(\mathbf{A}^{(l-1)}, \mathbf{X}^{(l-1)}\right).\tag{4}$$

These approaches differ from each other in the way to produce $\mathbf{S}$, which is used to inject a bias in the formation of clusters. In our work, we select three GP methods, i.e., DiffPool [50], MinCutPool [2], and DMoNPool [34], to cope with. Details of the pooling layers in these methods are summarized in Appendix A.

**Topological Features of Graphs.** A simplicial complex $K$ consists of a set of simplices of certain dimensions. Each simplex $\gamma \in K$ has a set of faces, and each face $\tau \in \gamma$ has to satisfy $\tau \in K$. An element $\gamma \in K$ with $|\gamma| = k + 1$ is called a $k$-simplex, which we denote by writing $\dim \gamma = k$. Furthermore, if $k$ is maximal among all simplices in $K$, then $K$ is referred to as a $k$-dimensional simplicial complex. A graph can be seen as a low-dimensional simplicial complex that only contains 0-simplices (vertices) and 1-simplices (edges) [19]. The simplest kind of topological features describing graphs are Betti numbers, formally denoted as $\beta_0$ for the number of connected components and $\beta_1$ for the number of cycles.

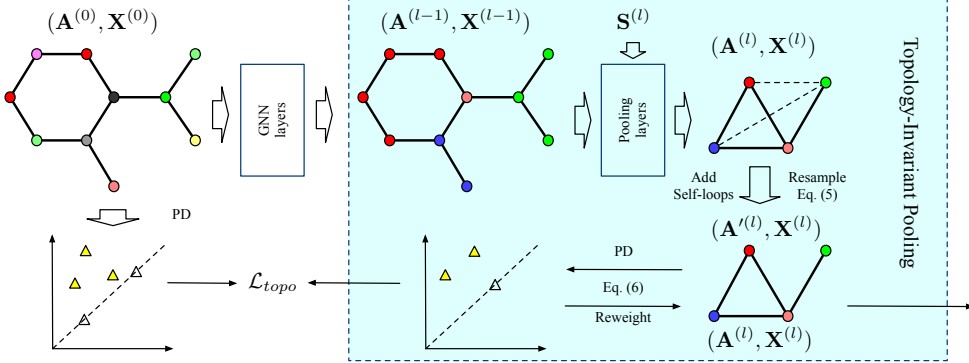

Figure 2: Overview of our method. The shaded part is one layer of Topology-Invariant Pooling.

Despite the limited expressive power of these two numbers, it can be improved by evaluating them alongside a filtration. Filtrations are scalar-valued functions of the form $f : V \cup E \to \mathbb{R}$. Changes in the Betti numbers, named as persistent Betti numbers, can subsequently be monitored throughout the progress of the filtration: by considering a threshold ($a \in \mathbb{R}$), we can analyze the subgraph originating from the pre-image of $((-\infty, a])$ of $f$, denoted as $(f^{-1}((-\infty, a]))$. The image of $f$ leads to a finite set of values $a_1 < \cdots < a_n$ and generates a sequence of nested subgraphs of the form $\emptyset \subseteq \mathcal{G}_0 \subseteq \ldots \mathcal{G}_k \ldots \subseteq \mathcal{G}_n = \mathcal{G}$, where $\mathcal{G}_k = (V_k, E_k)$ is a subgraph of $\mathcal{G}$ with $V_k := \{v \in V \mid f(\mathbf{x}_v) \leq a_k\}$ and $E_k := \{(v, w) \in E \mid \max\{f(x_v), f(x_w)\} \leq a_k\}$. This process is also known as persistent homology (denoted as $ph(\cdot)$) on graphs. Typically, persistent Betti numbers are summarized in a persistence diagram (PD) as $\mathrm{ph}(\mathcal{G}, f)[i] = \mathcal{D}_i$, where $i \in [0, 1, ...]$ is the dimension of topological features. PD is made up of tuples $(a_i, a_j) \in \mathbb{R}^2$, with $a_i$ and $a_j$ representing the creation and destruction of a topological feature respectively (see Fig. 1(c)). The absolute difference in function values $|a_j - a_i|$ is called the persistence of a topological feature, where high persistence corresponds to features of the function, while low persistence is typically considered as noise [19, 39].

## 4 Methodology

### 4.1 Overview

An overview of our method is shown in Fig. 2, where the shaded part corresponds to one layer of Topology-Invariant Pooling. The upper part is the GP process and the lower part is the injection of PH. Let $(\mathbf{A}^{(0)}, \mathbf{X}^{(0)})$ be the input graph. We consider to perform a GP at the $(l-1)$-th layer. After obtaining a coarsened (densely connected) graph $(\mathbf{A}^{(l)}, \mathbf{X}^l)$ with a standard GP method, we resample the coarsened graph using Gumbel-softmax trick as $\mathbf{A}'^{(l)}$ in order to make it adapt to PH. Then, this coarsened graph is further reweighted injecting persistence, and is optimized by minimizing the topological gap $\mathcal{L}_{topo}$ from the original graph, yielding $(\mathbf{A}^{(l)}, \mathbf{X}^l)$. By stacking multiple TIP layers, hierarchical pooling emphasizing topological information can be achieved. In the following sections, we elaborate on the detailed design of our mechanism.

### 4.2 Topology-Invariant Pooling

In many real-world applications, the topology of graphs are of utmost importance [44, 49, 16]. However, typical GNNs fail to capture certain topological structures in graphs, such as cycles [4, 51, 21]. Moreover, in dense graph pooling, graphs are pooled without preserving any topology. Even if we manage to make GNN topology-aware, the coarsened graph is nearly fully connected and has no meaningful topology at all, impairing the use of GNNs in these tasks. To overcome these limitations, we propose to inject topological information into GP. We resort to PH to characterize the importance of edges.

The core of PH is the notion of filtration, the selection of which presents a challenging task. As the coarsened graph evolves in each training step, integrating PH into GP demands multiple computations

of filtrations. To address this, we place recently proposed learnable filtration (LF) functions [16] to incorporate PH information for flexibility and efficiency. LF relies on node features and graph topology, which are readily available in GP. Consequently, LF can be seamlessly integrated into GP with minimal computational overhead. Specifically, we employ an MLP network $\Phi(\cdot)$ as the filtration function together with $\text{sigmoid}(\cdot)$ to map node features $\mathbf{X} \in \mathbb{R}^{n \times d}$ into $n$ scalar values. Recently, an increasing amount of attention has been devoted to cycles [4, 51, 21] due to their significant relevance to downstream tasks in various domains such as biology [26], chemistry [35], and social network analysis [23]. Recognizing that cycles offer an intuitive representation of graph structure [31, 17], and preliminary experiments, shown in Appendix E.5, indicate that the additional inclusion of zero-dimensional topological features merely increases runtime, thus we instead focus on the one-dimensional PDs associated with cycles. For those edges do not form cycles, their creation and destruction are the same, leading to zero persistence. Following the standard way in GP (Eq. 2 3 4), we additionally propose the subsequent modules to inject PH into GP at both feature and topology levels.

**Resampling.**    One major limitation of utilizing LF proposed in [16] is that the computation process is unaware of edge weights, i.e. edges with non-zero weights will be treated equally, so PH cannot directly extract meaningful topology from $\mathbf{A}^{(l)}$. Besides, rethinking GP in Eq. 3, the coarsened adjacency matrix has limited expressive power for two reasons. First, although $\mathbf{S}^{(l)}$ is a soft assignment matrix obtained by $\text{softmax}(\cdot)$, each element still has nonzero values, i.e. $\mathbf{A}^{(l)}$ is always densely connected. Second, the edge weights may span a wide range by multiplication (refer to Appendix D for empirical evidence). These drawbacks hinder the stability and generalization power of the subsequent message passing layers [13]. None of the existing GP methods can handle these problems properly.

Therefore, we resample the coarsened adjacency $\mathbf{A}^{(l)}$ obtained from a normal GP layer (Eq. 3) as:

$$\mathbf{A}'^{(l)} = \text{resample}\left(\frac{\mathbf{A}^{(l)} - \min(\mathbf{A}^{(l)})}{\max(\mathbf{A}^{(l)}) - \min(\mathbf{A}^{(l)})}\right), \tag{5}$$

where $\mathbf{A}^{(l)}$ is first normalized in the range of $[0, 1]$, and $\text{resample}(\cdot)$ is performed independently for each matrix entry using the Gumbel-softmax trick [22]. In practice, only the upper triangular matrix is resampled to make it symmetric and we add self-loops to the graph.

**Persistence Injection.**    Now $\mathbf{A}'^{(l)} \in \{0, 1\}^{n_l \times n_l}$ is a sparse matrix without edge features so we can easily inject topological information into it. For a resampled graph with $\mathbf{A}'^{(l)}$ and $\mathbf{X}^{(l)}$, we formulate the persistence injection as:

$$\begin{aligned}
\tilde{\mathcal{D}}_1 &= \text{ph}(\mathbf{A}'^{(l)}, \text{sigmoid}(\Phi(\mathbf{X}^{(l)})))[1] \\
\mathbf{A}^{(l)} &= \mathbf{A}'^{(l)} \odot \text{to\_dense}(\tilde{\mathcal{D}}_1[1] - \tilde{\mathcal{D}}_1[0]),
\end{aligned} \tag{6}$$

where $\odot$ is the Hadamard product, $\text{to\_dense}()$ means transforming sparse representations in terms of edges to dense matrix representations, $\tilde{\mathcal{D}}_1$ is the augmented 1-dimensional PDs by placing the tuples correspond to self-loop edges on the diagonal part of original PDs $\mathcal{D}_1$, $\tilde{\mathcal{D}}_1[i]$ is the $i$-th value in each tuple of $\tilde{\mathcal{D}}_1$, and we denote the updated adjacency matrix after persistence injection still as $\mathbf{A}^{(l)}$ for notation consistency. Persistence injection can actually be regarded as a reweighting process. Since the filtration values are within $[0, 1]$, $\mathbf{A}^{(l)}$ after persistence injection is guaranteed to have edge weights in the range of $[0, 1]$ and is passed to the next pooling layer.

**Topological Loss Function.**    The aforementioned mechanism can explicitly inject topological information into graphs, but it relies on the condition that the coarsened graph retains certain essential sub-topology. To this end, we propose an additional loss function to guide the GP process.

Intuitively, the coarsened graph should exhibit similarity to the original graph in terms of topology. Since the computation of PH is differentiable, one possible approach is to directly minimize the differences between the PDs of the original graph and the coarsened graph. However, this implementation would require computing the Wasserstein distance between two PDs through optimal transport [48], which is intractable in training due to its complexity. Considering that our objective is to estimate the difference, we instead propose vectorizing the PDs and minimizing their high-order statistical

features [36]. Specifically, we use several transformations (denoted as $\text{transform}(\cdot)$) and concatenate the output, including triangle point transformation, Gaussian point transformation and line point transformation introduced in [6] to convert the tuples in PD into vector $\mathbf{h}_t$ ($t \in [1, m]$). We calculate the mean vector $\mu$ as well as the second-order statistics as the standard deviation vector $\sigma$ as:

$$\mathbf{h}_t = \text{transform}(\tilde{\mathcal{D}}_1)$$

$$\mu = \frac{1}{m} \sum_{t=1}^{m} \mathbf{h}_t, \qquad \sigma = \sqrt{\frac{1}{m} \sum_{t=1}^{m} \mathbf{h}_t \odot \mathbf{h}_t - \mu \odot \mu} \tag{7}$$

In this manner, the difference between PDs can be estimated through the comparison of their statistics in the features, which is the concatenation of the mean and variance vectors. To further regularize the topological difference between layers, we introduce a topological loss term defined as:

$$\mathcal{L}_{topo} = \frac{1}{Ld} \sum_{l=1}^{L} \sum_{i=1}^{d} \left( \left( \mu_i^{(l)} \| \sigma_i^{(l)} \right) - \left( \mu_i^{(0)} \| \sigma_i^{(0)} \right) \right)^2, \tag{8}$$

where $(\cdot \| \cdot)$ stands for the concatenation operation, $L$ is the number of pooling layers, and $d$ is the feature dimension. Note that the intuition behind $\mathcal{L}_{topo}$ is different from the loss functions in existing graph pooling methods: the coarsened graph after pooling should be topologically similar to the original graph rather than having exact cluster structures.

## 4.3 Analysis

In this section, we examine the validity of our proposed method, and in particular, analyze its expressive power and complexity.

**Theorem 1.** *The self-loop augmented 1-dimensional topological features computed by PH is sufficient enough to be at least as expressive as 1-WL in terms of distinguishing non-isomorphic graphs, i.e. if the 1-WL label sequences for two graphs $\mathcal{G}$ and $\mathcal{G}'$ diverge, there exists an injective filtration $f$ such that the corresponding 1-dimensional persistence diagrams $\tilde{\mathcal{D}}_1$ and $\tilde{\mathcal{D}}_1'$ are not equal.*

**Proof Sketch.** We first assume the existence of a sequence of WL labels and show how to construct a filtration function $f$ from this. Consider nodes $u$ and $u'$ are nodes with unique label count in $\mathcal{G}$ and $\mathcal{G}'$, then our filtration is constructed such that their filtration values $f(u)$ and $f(u')$ are unique and different. Consider all three cases: (1) $u$ and $u'$ are both in cycles; (2) $u$ and $u'$ are both not in cycles; (3) one of $u$ and $u'$ is in cycles and the other is not. For all the cases, $f(u)$ and $f(u')$ will be revealed in their respective persistence diagrams. Since $f(u)$ and $f(u')$ are unique and different, we can use the augmented persistence diagrams to distinguish the two graphs.

This result demonstrates that the self-loop augmented 1-dimensional topological features contain sufficient information to potentially perform at least as well as 1-WL when it comes to distinguishing non-isomorphic graphs. We can then obtain the concluding remark that TIP is more expressive than other dense pooling methods by showing that there are pairs of graphs that cannot be distinguished by 1-WL but can be distinguished by TIP. Besides, our proposed simple yet effective self-loop augmentation eliminates the necessity of computing 0-dimensional topological features, thus reducing computational burdens.

**Proposition 1.** *TIP is invariant under isomorphism.*

Detailed proof and illustrations of the theorem and proposition can be found in Appendix C.

**Complexity.** PH can be efficiently computed for dimensions 0 and 1, with a worst-case time complexity of $O(m\alpha(m))$, where $m$ represents the number of sorted edges in a graph. Here, $\alpha(\cdot)$ represents the inverse Ackermann function, which is extremely slow-growing and can essentially be considered as a constant for practical purposes. Therefore, the primary factor that affects the calculation of PH is the complexity of sorting all the edges, which is $O(m \log m)$. Our resampling and persistence injection mechanism ensures that the coarsened graphs are sparse rather than dense, making our approach both efficient and scalable. We provide running time comparisons in Appendix E.2, which indicates that the inclusion of TIP does not impose a significant computational burden.

# 5 Experiments

In the experiments, we evaluate the benefits of persistent homology on several state-of-the-art graph pooling methods, with the goal of answering the following questions:

**Q1.** Is PH capable of preserving topological information during pooling?

**Q2.** How does PH affect graph pooling in preserving task-specific information?

To this end, we showcase the empirical performance of TIP on two tasks, namely, topological similarity (Section 5.2) and graph classification (5.3). Our primary focus is to assess in which scenarios topology can enhance GP.

## 5.1 Experimental Setup

**Models.** To investigate the effectiveness of PH in GP, we integrate TIP with DiffPool, MinCutPool, and DMoNPool, which are the pioneering approaches that have inspired many other pooling methods. Additionally, as most pooling methods rely on GNNs as their backbone, we compare the widely used GNN models GCN [25], GIN [47], and GraphSAGE [15]. We also look into another two related and State-of-the-Art GNN models, namely TOGL [19] and GSN [4], which incorporate topological information and graph substructures into GNNs to enhance the expressive power. Several other GP methods, namely Graclus [7] and TopK [11] are also compared. For model selection, we follow the guidelines provided by the original authors or benchmarking papers. Our method acts as an additional plug-in to existing pooling methods (referred to as -TIP) without modifying the remaining model structure and hyperparameters. Appendix B.1 provides detailed configurations of these models.

**Datasets.** To evaluate the capabilities of our model across diverse domains, we assess its performance on a variety of graph datasets commonly used in graph related tasks. We select several benchmarks from TU datasets [32], OGB datasets [20] and ZINC dataset [43]. Specifically, we adopt molecular datasets NCI1, NCI109, and OGBG-MOLHIV, bioinformatics datasets ENZYMES, PROTEINS, and DD, as well as social network datasets IMDB-BINARY and IMDB-MULTI. Furthermore, to investigate the topology-preserving ability of our method, we conduct experiments on several highly structured datasets (ring, torus, grid2d) obtained from the PyGSP library. Appendix B.2 provides detailed statistics of the datasets.

**Evaluation.** In the graph classification task, all datasets are splitted into train (80%), validation (10%), and test (10%) data. Following the evaluation protocol in [50, 30], we train all models using the Adam optimizer [24] and implement a learning rate decay mechanism, reducing the learning rate from $10^{-3}$ to $10^{-5}$ with a decay ratio of 0.5 and a patience of 10 epochs. Additionally, we use early stopping based on the validation accuracy with patience of 50 epochs. We report statistics of the performance metrics over 20 runs with different seeds.

## 5.2 Preserving Topological Structure

In this experiment, we study **Q1** about the ability of PH to preserve topological structure during pooling. Specifically, we assess the topological similarity between the original and coarsened graphs $\mathcal{G}$ and $\mathcal{G}'$, by comparing the Wasserstein distance associated with their respective PDs $\tilde{\mathcal{D}}_1$ and $\tilde{\mathcal{D}}'_1$. This evaluation criterion is widely used to compare the topological similarity of graphs [48, 41]. We utilize Forman curvature on each edge of the graph as the filtration, which incorporates edge weights and graph clusters to better capture the topological features of the coarsened graphs [42, 45]. We consider the 1-Wasserstein distance $W\left(\tilde{\mathcal{D}}_1, \tilde{\mathcal{D}}'_1\right) = \inf_{\delta \in \Pi\left(\tilde{\mathcal{D}}_1, \tilde{\mathcal{D}}'_1\right)} \mathbb{E}_{(x,y)\sim\delta}[\|x - y\|]$ as the evaluation metric, where $\Pi(\cdot)$ is the set of joint distributions $\delta(x, y)$ whose marginals are $\tilde{\mathcal{D}}_1$ and $\tilde{\mathcal{D}}'_1$, respectively. Note that we are not learning a new filtration but keep a fixed one. Rather, we use learnable filtrations in training to enhance flexibility, and solely optimize $L_{topo}$ as the main objective.

We compare TIP with other pooling methods. Table 1 reports the average $W$ values on three datasets, demonstrating that TIP can improve dense pooling methods to a large margin and have the best topological similarity. We visualize the pooling results in Fig. 3 for better interpretation, where isolated nodes with no links are omitted for clarity. It is evident that DiffPool, MinCutPool, and

Table 1: Results to show the topology-preserving ability. Wasserstein distance (↓) is used to assess the topological similarity. A **bold** value indicates the overall winner.

| Methods | Datasets | | |
|---|---|---|---|
| | ring | torus | grid2d |
| Graclus | $37.62 \pm 4.41$ | $124.47 \pm 12.07$ | $35.82 \pm 0.93$ |
| TopK | $14.24 \pm 1.06$ | $35.15 \pm 4.78$ | $84.12 \pm 2.21$ |
| DiffPool | $234.57 \pm 9.49$ | $237.89 \pm 20.66$ | $146.91 \pm 6.05$ |
| DiffPool-TIP | $\mathbf{8.03 \pm 3.08}$ | $17.97 \pm 2.19$ | $\mathbf{32.26 \pm 3.21}$ |
| MinCutPool | $232.60 \pm 10.81$ | $248.51 \pm 15.69$ | $155.16 \pm 21.79$ |
| MinCutPool-TIP | $18.11 \pm 5.59$ | $\mathbf{11.38 \pm 2.21}$ | $58.71 \pm 9.84$ |
| DMoNPool | $224.48 \pm 22.25$ | $236.97 \pm 16.54$ | $142.85 \pm 27.53$ |
| DMoNPool-TIP | $16.10 \pm 4.80$ | $17.34 \pm 4.76$ | $52.26 \pm 5.75$ |

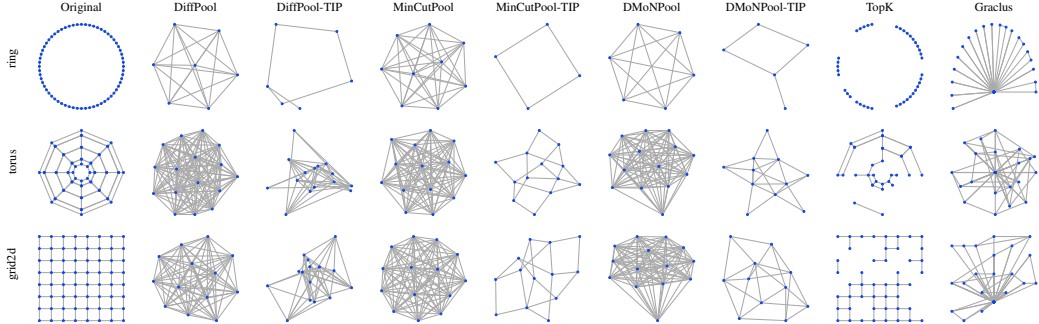

Figure 3: Coarsened graphs from different methods in the preserving topological structure experiment.

DMoNPool tend to generate dense graphs and fail to preserve any topological structures. Conversely, our method, which incorporates topological features using PH, sparsifies the coarsened graphs and reveals certain essential topological structures. Notably, in the ring and torus datasets, large cycles are clearly preserved by our method. Besides, the grid2d dataset, despite having a different spatial layout, exhibits similar topology to torus (with four adjacent nodes forming a small cycle), resulting in similar shapes of their corresponding coarsened graphs. This indicates that the objective function indeed contributes to preserving topological similarity to some extent. Sparse pooling methods, which tend to preserve local topology, perform slightly better than the original dense pooling methods.

Table 2: Test accuracy (↑) of graph classification on benchmark datasets. A **bold** value indicates the overall winner. Gray background indicates that TIP outperforms the base GP.

| Methods | Datasets | | | | | | | |
|---|---|---|---|---|---|---|---|---|
| | NCI1 | NCI109 | ENZYMES | PROTEINS | DD | IMDB-BINARY | IMDB-MULTI | OGBG-MOLHIV |
| GCN | $77.81 \pm 1.50$ | $74.90 \pm 1.85$ | $32.51 \pm 3.35$ | $76.65 \pm 3.14$ | $78.66 \pm 2.36$ | $74.20 \pm 2.40$ | $53.23 \pm 3.04$ | $75.04 \pm 0.84$ |
| GIN | $80.30 \pm 1.70$ | $79.66 \pm 1.55$ | $42.83 \pm 3.66$ | $77.18 \pm 3.35$ | $78.05 \pm 3.60$ | $72.65 \pm 3.04$ | $53.28 \pm 3.16$ | $76.03 \pm 0.84$ |
| GraphSAGE | $80.85 \pm 1.25$ | $79.16 \pm 1.28$ | $39.17 \pm 3.28$ | $76.67 \pm 3.05$ | $78.83 \pm 3.07$ | $76.60 \pm 2.37$ | $53.46 \pm 2.39$ | $76.18 \pm 1.27$ |
| TOGL | $80.53 \pm 2.29$ | $78.27 \pm 1.39$ | $46.09 \pm 3.72$ | $78.17 \pm 2.80$ | $76.10 \pm 2.24$ | $76.65 \pm 2.75$ | $53.87 \pm 2.67$ | $77.21 \pm 1.33$ |
| GSN | $83.50 \pm 2.00$ | $79.45 \pm 1.88$ | $49.50 \pm 6.54$ | $74.59 \pm 5.00$ | $73.17 \pm 4.17$ | $\mathbf{76.80 \pm 2.00}$ | $52.60 \pm 3.60$ | $76.06 \pm 1.74$ |
| Graclus | $80.82 \pm 1.27$ | $79.13 \pm 1.79$ | $41.44 \pm 3.46$ | $75.69 \pm 2.62$ | $74.67 \pm 2.45$ | $74.45 \pm 3.29$ | $54.72 \pm 2.79$ | $76.81 \pm 0.70$ |
| TopK | $79.43 \pm 3.50$ | $77.96 \pm 1.58$ | $38.35 \pm 4.83$ | $76.03 \pm 2.94$ | $76.97 \pm 3.94$ | $72.60 \pm 4.24$ | $53.66 \pm 2.93$ | $76.28 \pm 0.67$ |
| DiffPool | $77.64 \pm 1.86$ | $76.50 \pm 2.32$ | $48.34 \pm 5.14$ | $78.81 \pm 3.12$ | $80.27 \pm 2.51$ | $73.15 \pm 3.30$ | $54.32 \pm 2.99$ | $76.60 \pm 1.04$ |
| DiffPool-TIP | $\mathbf{83.75 \pm 1.31}$ | $\mathbf{81.09 \pm 1.65}$ | $\mathbf{65.05 \pm 4.24}$ | $\mathbf{79.86 \pm 3.12}$ | $\mathbf{82.12 \pm 2.53}$ | $76.40 \pm 3.13$ | $\mathbf{55.53 \pm 2.92}$ | $\mathbf{77.75 \pm 1.18}$ |
| MinCutPool | $77.92 \pm 1.67$ | $75.88 \pm 2.06$ | $39.83 \pm 2.63$ | $78.25 \pm 3.84$ | $79.15 \pm 3.51$ | $73.80 \pm 3.54$ | $53.87 \pm 2.95$ | $75.60 \pm 0.54$ |
| MinCutPool-TIP | $80.17 \pm 1.29$ | $79.48 \pm 1.37$ | $46.34 \pm 3.85$ | $79.73 \pm 3.27$ | $80.87 \pm 2.47$ | $75.20 \pm 2.67$ | $54.47 \pm 2.27$ | $77.18 \pm 0.83$ |
| DMoNPool | $78.03 \pm 1.64$ | $76.62 \pm 1.94$ | $40.82 \pm 3.68$ | $78.63 \pm 3.89$ | $79.16 \pm 3.61$ | $73.50 \pm 3.01$ | $54.07 \pm 3.08$ | $76.30 \pm 1.34$ |
| DMoNPool-TIP | $79.68 \pm 1.38$ | $78.46 \pm 1.50$ | $45.84 \pm 5.32$ | $79.73 \pm 3.66$ | $81.46 \pm 2.96$ | $74.25 \pm 2.93$ | $54.23 \pm 2.64$ | $76.70 \pm 0.62$ |

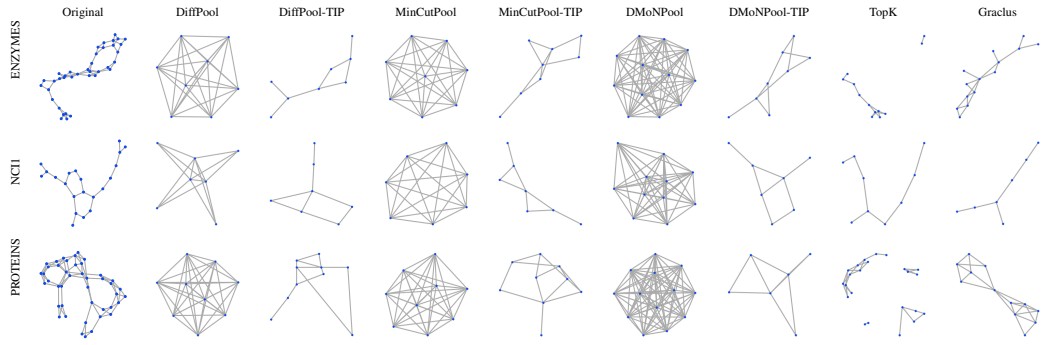

Figure 4: Graphs pooled with different methods in graph classification experiment.

## 5.3 Preserving Task-Specific Information

In this experiment, we examine the impact of PH on GP in downstream tasks to answer **Q2**. We have observed in the former experiment that PH can preserve essential topological information during pooling. However, two additional concerns arise: (1) Does TIP continue to generate invariant sub-topology in the downstream task? (2) If so, does this sub-topology contribute to the performance of the downstream task? To address these concerns, we evaluate TIP using various graph classification benchmarks, where the accuracy achieved on these benchmarks serves as a measure of a method's ability to selectively preserve crucial information based on the task at hand.

We begin by visualizing the coarsened graphs in this task, where edges are cut-off by a small value. From Fig. 4, we can clearly observe that our method manage to preserve the essential sub-topology similar to the original graphs, while dense pooling methods cannot preserve any topology. As discussed in [30], dense pooling methods achieve comparable performance when the assignment matrix **S** is replaced by a random matrix. Here our visualization reveals that regardless of the value of **S**, the coarsened graph always approaches to a fully connected one. Sparse pooling methods, on the other hand, manage to preserve some local structures through clustering or dropping, but the essential global topological structures are destroyed.

Table 2 presents the average and standard deviation of the graph classification accuracy on benchmark datasets, where the results of GP and several baseline GNNs are provided. Experimental results demonstrate that TIP can consistently enhance the performance of the three dense pooling methods. While the original dense pooling methods sometimes underperform compared to the baselines, they are able to surpass them after integrating TIP.

Moreover, an intriguing observation can be found on ENZYMES dataset, where TOGL surpasses the baseline GNNs. TOGL in practice, incorporates PH into GNNs (GraphSAGE in our implementation), so this results underscores the significance of incorporating topological information for improved performance on ENZYMES. Further, our method demonstrates more significant improvements by augmenting the three dense pooling methods on the ENZYMES dataset. One possible explanation for the observed phenomenon is that the coarsened graphs generated by our methods bear a striking resemblance to numerous frequent subgraphs present in this dataset [10]. Such substructures may serve as indicators of unique characteristics within the graph, rendering them valuable for subsequent tasks. However, it is also worth noting that TOGL only exhibits marginal improvements or even underperforms on the other datasets. This suggests that simply integrating PH features into GNN layers does not fully exploit topological information. Conversely, injecting global topological invariance into pooling layers in our method yields superior performance.

To demonstrate the effectiveness of preserving the invariant sub-topology, we compared DiffPool-TIP with its variant counterpart, DiffPool-TIP-NL (no topological loss), by replacing $\mathcal{L}_{topo}$ with the original $\mathcal{L}_r$ in DiffPool (see Table 4 in Appendix A). The training objective curve and the Wasserstein distance curve are presented in Figure 5, both based on the ENZYMES dataset and a fixed filtration (the same as in Section 5.2). From the figures, it is evident that the objective value decreases as the coarsened graphs become more similar in topology to the original graphs when using DiffPool-TIP. However, when training without $\mathcal{L}_{topo}$, the performance is inferior. Additionally, even when the

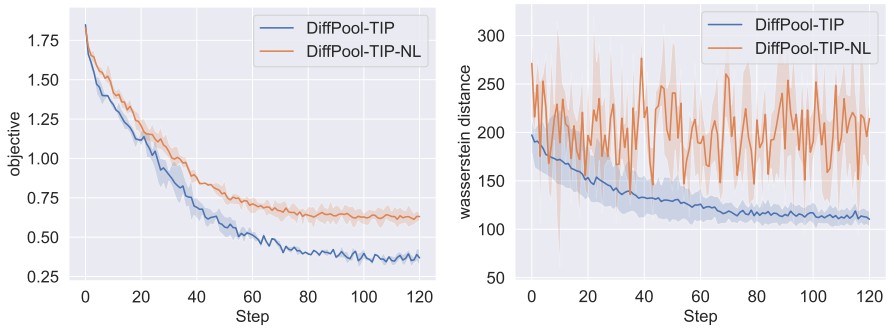

Figure 5: The training curves of DiffPool-TIP and DiffPool-TIP-NL on ENZYMES dataset. We show the average values and min-max range of objective and Wasserstein distance for multiple runs.

objective value converges, DiffPool-TIP-NL still exhibits changing topology, whereas DiffPool-TIP maintains a stable topology, possibly benefiting from the stability of PH [41]. This also suggests that multiple suboptimal topologies may contribute equally to the objective. Our topology invariant pooling strategy consistently selects topologies similar to the original graph, which leads to better performance. Additional visualization results and analysis about the coarsened graphs obtained by DiffPool-TIP-NL can be found in Appendix E.3.

Aside from the graph classification task, Table 3 presents the mean and standard deviation of prediction accuracy for the constrained solubility of molecules in the ZINC dataset, where mean square error is used as performance metric. We can observe that TIP can still boost the three pooling methods on regression task, which demonstrates that our proposed method can retain task-related information. Besides, we design an additional set of experiments in Appendix E.4, where the topological structure of the graph is highly task-relevant. **Ablation study** about the contributions of different modules are shown in Appendix E.5. Finally, to empirically demonstrate the expressive power of our proposed method, we provide an experiment on distinguishing non-isomorphic graphs in Appendix E.6.

Table 3: Mean square error ($\downarrow$) of prediction results on ZINC dataset. A **bold** value indicates the overall winner.

|  | ZINC |
| --- | --- |
| DiffPool | 0.34±0.01 |
| DiffPool-TIP | **0.28±0.01** |
| MinCutPool | 0.42±0.01 |
| MinCutPool-TIP | **0.38±0.01** |
| DMoNPool | 0.40±0.01 |
| DMoNPool-TIP | **0.35±0.01** |

## 6 Conclusion

In this paper, we developed a method named Topology-Invariant Pooling (TIP) that effectively integrates global topological invariance into graph pooling layers. This approach is inspired by the observation that the filtration operation in PH naturally aligns with the GP process. We theoretically showed that PH is at least as expressive as WL-test, with evident examples demonstrating TIP's expressivity beyond dense pooling methods. Empirically, TIP indeed preserved persistent global topology information, and achieved substantial performance improvement on top of several pooling methods on various datasets, demonstrating strong flexibility and applicability.

The potential limitation of our study is the heavy reliance of the proposed method on circular structures within graphs, potentially hindering its efficacy on tree-like graphs. Besides, our method lacks the ability to discriminate between graphs when the number of connected components is the only distinguishing factor. Our method can be extended to address this limitation by explicitly incorporating this information into the node features during the pooling process.

## Acknowledgements

This work was supported by the National Key R&D Program of China under grant 2022YFA1003900. This work is also supported by the Guangdong Provincial Key Laboratory of Mathematical Foundations for Artificial Intelligence (2023B1212010001).

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

Table 4: Unsupervised loss functions of graph pooling

| Method | $\mathcal{L}_r$ | $\mathcal{L}_c$ |
|---|---|---|
| DiffPool | $\left\| \mathbf{A}, \mathbf{SS}^T \right\|_F$ | $\frac{1}{n}\sum_{i=1}^{n} H\left(\mathbf{S}_i\right)$ |
| MinCutPool | $-\frac{\text{Tr}\left(\mathbf{S}^\top \mathbf{AS}\right)}{\text{Tr}\left(\mathbf{S}^\top \mathbf{DS}\right)}$ | $\left\| \frac{\mathbf{S}^\top \mathbf{S}}{\|\mathbf{S}^\top \mathbf{S}\|_F} - \frac{\mathbf{I}_C}{\sqrt{C}} \right\|_F$ |
| DMoNPool | $-\frac{1}{2m} \cdot \text{Tr}\left(\mathbf{S}^\top \mathbf{BS}\right)$ | $\left\| \frac{\mathbf{S}^\top \mathbf{S}}{\|\mathbf{S}^\top \mathbf{S}\|_F} - \frac{\mathbf{I}_C}{\sqrt{C}} \right\|_F + \frac{\sqrt{C}}{n} \left\| \sum_i \mathbf{S}_i^\top \right\|_F - 1$ |

## A  Dense Graph Pooling Methods

Generally, dense graph pooling methods follow a hierarchical architecture, but their motivations differ. DiffPool suggests that nearby nodes should be pooled together, drawing on insights from link prediction and the assignment matrix $\mathbf{S}$ should be approximate to a one-hot vector so that the clusters are less overlapped with each other. MinCutPool, on the other hand, adapts the normalized cut as a regularizer for pooling. This encourages strongly connected nodes to be pooled together, ensures orthogonal cluster assignments, and promotes clusters of similar size. Moreover, DMoNPool additionally proposes a regularization to optimize the modularity quality of clusters so that the pooling can generate high quality clusters approach to ground truth. In summary, each of these methods introduces two types of unsupervised loss functions: the reconstruction loss $\mathcal{L}_r$, which regulates how the coarsened graph is reconstructed to retain some cluster structure, and the other is the cluster loss $\mathcal{L}_c$, which prevents convergence to local minima. The detailed formulations of these loss functions are provided in Table 4, where $|| \cdot ||_F$ denotes the Frobenius norm, $H$ denotes the entropy function, $\mathbf{S}_i$ is the $i$-th row of $\mathbf{S}$, $\mathbf{D}$ is the degree matrix, $C$ is the number of clusters, $\mathbf{B} = \mathbf{A} - \frac{\mathbf{D}\mathbf{D}^T}{2m}$ is the modularity matrix, respectively.

## B  Experimental Setup

### B.1  Implementation detail

**Hyperparameters.**  For dense pooling methods, the pooling ratio ranges from $[0.1, 0.5]$, the number of pooling layers is 2, and the hidden dimension is selected from $\{32, 64\}$. For the Graclus method we use 2 pooling layers, while for TopK we use 3 pooling layers with a pooling ratio of 0.8. The batch size for all models is uniformly set to 20, and the maximum number of training epochs is 1000. For the graphs obtained from the PyGSP library (ring, torus, grid2d), the number of nodes in each graph is fixed at 64.

**Model configuration.**  All the methods are implemented using PyTorch and PyG [37, 9]. The compared methods are implemented following the implementations provided in the PyG library [2]. In the case of DiffPool, it uses a 3-layer GraphSAGE in each pooling layer, while MinCutPool and DMoNPool use a 1-layer GCN before pooling and a 1-layer GNN [33] in each pooling layer. Note that in DiffPool, the GNNs in Eqs. 2 and 4 are different, while in MinCut and DNoNPool they are the same one, as what their do in the original papers. TopK and Graclus are based on a 1-layer GNN [33]. TOGL is implemented using a 3-layer GraphSAGE as it has demonstrated superior performance on graph classification tasks (see Table 2). For the baseline GNN models (GCN, GIN, and GraphSAGE), we use 3 layers with mean/max pooling. In our model, TIP is incorporated as a plugin to existing pooling methods, without modifying the remaining model structure and hyperparameters. We replace the reconstruction loss $\mathcal{L}_r$ with $\mathcal{L}_{topo}$ while keeping the cluster loss $\mathcal{L}_c$ unchanged. In the case of MinCutPool and DMoNPool, our resampling strategy is added after their original normalization of the coarsened graphs. In preserving topological structure experiments, we initialize node features as the concatenation of the first ten eigenvectors of graph Laplacian matrices. Moreover, we follow the settings in previous works [19, 18] to extend $\widehat{\mathcal{D}}_1$ as follows: (1) each cycle is paired with the edge that created it; (2) edges $e$ that do not create a cycle (still in this circle) are assigned a 'dummy' tuple value, such as $(f(e), f(e))$; (3) all other edges will be paired with the maximum value of the filtration $f_{max}$. In practice we set $f_{max}$ plus a constant as infinity

---

[2] https://pytorch-geometric.readthedocs.io/en/latest/modules/nn.html

Table 5: Statistics of datasets

| Dataset | #Graphs | #Avg.Nodes | #Avg.Edges | #Features | #Classes |
|---------|---------|------------|------------|-----------|----------|
| ENZYMES | 600 | 32.63 | 62.14 | 18 | 6 |
| PROTEINS | 1113 | 39.06 | 72.82 | 3 | 2 |
| NCI1 | 4110 | 29.87 | 32.30 | 37 | 2 |
| NCI109 | 4127 | 29.68 | 32.13 | 38 | 2 |
| DD | 1060 | 232.9 | 583 | 89 | 2 |
| IMDB-BINARY | 1000 | 19.8 | 193.1 | 0 | 2 |
| IMDB-MULTI | 1500 | 13 | 65.94 | 0 | 3 |
| OGBG-MOLHIV | 41127 | 25.5 | 27.5 | 9 | 2 |
| ZINC | 249456 | 23.2 | 49.8 | 1 | 1 |

of destruction time. Therefore, $\tilde{\mathcal{D}}_1$ consists of as many tuples as the number of edges $m$. Code is open-sourced at `https://github.com/LOGO-CUHKSZ/TIP.git`.

## B.2 Dataset Statistics

The statistics of datasets used in this paper are summarized in Table 5, where we show the number of graphs, average number of nodes, average number of edges, number of features, and number of classes. We use the default dataset settings from PyG library [3]. Highly structured datasets (ring, torus, grid2d) are obtained from the PyGSP library [4].

## C Theoretical Expressivity of TIP

**Theorem 1.** *The self-loop augmented 1-dimensional topological features computed by PH is sufficient enough to be at least as expressive as 1-WL in terms of distinguishing non-isomorphic graphs, i.e. if the 1-WL label sequences for two graphs $\mathcal{G}$ and $\mathcal{G}'$ diverge, there exists an injective filtration $f$ such that the corresponding 1-dimensional persistence diagrams $\tilde{\mathcal{D}}_1$ and $\tilde{\mathcal{D}}'_1$ are not equal.*

*Proof.* Assume that $\mathcal{G}$ and $\mathcal{G}'$ have $n$ and $n'$ nodes, and the label sequences of them diverge at some iteration $h$, which means there exists at least one label whose count is unique. Let nodes $u$ and $u'$ be the nodes with unique count in $\mathcal{G}$ and $\mathcal{G}'$, respectively. Denote $La^{(h)} := \{l_1, l_2, ...\}$ as an enumeration of the finitely many hashed labels at iteration $h$. We can build a filtration function $f$ by assigning a vertex $v$ with label $l_i$ to its index, i.e. $f(v) := i$ except that $f(u) = n + n' + 1$ and $f(u') = n + n' + 2$. The filtration of edge $(u, w)$ is defined as $f(v, w) := max\{f(v), f(w)\}$, and for isolated nodes $v$, the filtration of self-loop edges is $f(v, v) = f(v)$. Therefore, node with unique label count and its connected edges always correspond to the largest filtration value. Note that the 1-dimensional PD has been extended to have the same cardinality as the number of edges. If node $u$ or $u'$ forms a circle, the creation of this circle is related to the edge with the largest filtration; if node $u$ or $u'$ does not form a circle, the corresponding edges lie on the diagonal of $\tilde{\mathcal{D}}_1$ with unique coordinates; otherwise node $u$ constitute a circle while $u'$ does not, then the corresponding edges lie in different parts in $\tilde{\mathcal{D}}_1$ and $\tilde{\mathcal{D}}'_1$. Hence, $\tilde{\mathcal{D}}_1 \neq \tilde{\mathcal{D}}'_1$.

To demonstrate that TIP is more expressive than other dense pooling methods, we provide examples of graph pairs that cannot be distinguished by 1-WL but can be by TIP. We present an example of such non-isomorphic graphs in Fig. 6, where in the second graph the edge connecting two triangles does not form a circle. This edge corresponds to zero persistence and is eliminated in TIP. Consequently, the two originally non-isomorphic graphs can be easily distinguished. Provided that the three sufficient conditions proposed in [3] are satisfied, the pooling layers retain the same level of expressive power as GNN. In TIP, the reduction of node features remains unaltered, thereby fulfilling the three conditions. Additionally, TIP is capable of distinguishing certain non-isomorphic graphs,

---

[3] `https://pytorch-geometric.readthedocs.io/en/latest/modules/datasets.html`
[4] `https://pygsp.readthedocs.io/en/stable/reference/graphs.html`

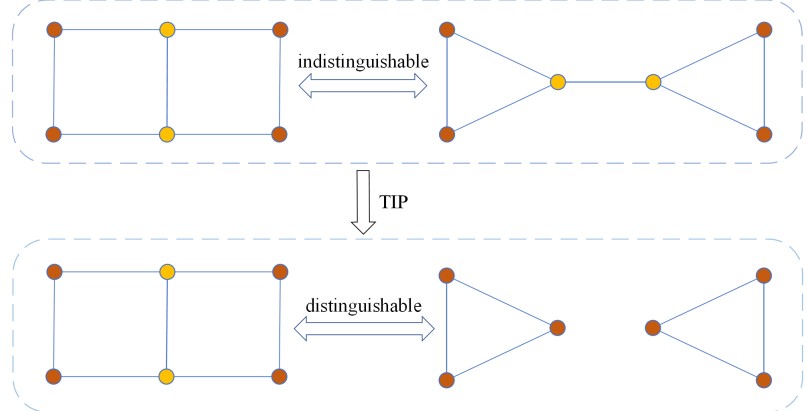

Figure 6: A pair of non-isomorphic graphs that cannot be distinguished by 1-WL but can be distinguished by TIP.

indicating its superior expressive power compared to conventional dense pooling methods such as DiffPool, MinCutPool, and DMoNPool.

$\square$

**Proposition 1.** *TIP is invariant under isomorphism.*

To prove this statement, we adopt the following lemma [39] to show the isomorphic property of PH.

**Lemma 1.** *Let $\mathcal{G}_1$ and $\mathcal{G}_2$ be two isomorphic graphs. For any equivariant filtration $f$, the corresponding persistence diagrams are equal.*

In TIP, the filtration $f$ is implemented using MLP, ensuring the equivariant property of filtration. Moreover, our resampling operations in Section 4.2 are equivariant. Therefore, the two isomorphic graphs after the resampling and persistence injection operations are still isomorphic to each other. Now we are able to prove Proposition 1.

*Proof.* For **feature-level invariance**, let $\mathbf{X} \in \mathbb{R}^{n \times d}$ be the node features, $\mathbf{P} \in \{0, 1\}^{n \times n}$ be the permutation matrix, $\mathbf{S} \in \mathbb{R}^{n \times n'}$ be the assignment matrix, and $\mathbf{PX}$ be the permutated node features. The node feature map after pooling is denoted as $\mathbf{X}' \in \mathbb{R}^{n' \times d}$, then we have $\mathbf{X}' = \mathbf{S}^\top \mathbf{X}$.

If we permute $\mathcal{G}$ using a permutation matrix $\mathbf{P}$, the permutated node features after pooling are

$$\mathbf{X}' = (\mathbf{S}^\top \mathbf{P}^\top)(\mathbf{PX}) = \mathbf{S}^\top \mathbf{X},$$

which proves the isomorphism invariant property of pooling at feature level.

For **connectivity-level invariance**, the connectivity after pooling is denoted as $\mathbf{A}' \in \mathbb{R}^{n' \times n'}$, then we have $\mathbf{A}' = \mathbf{S}^\top \mathbf{A} \mathbf{S}$. If we permute $\mathcal{G}$ using a permutation matrix $\mathbf{P}$, the permutated connectivity after pooling is

$$\mathbf{A}' = (\mathbf{S}^\top \mathbf{P}^\top)(\mathbf{PAP}^\top)(\mathbf{PS}) = \mathbf{S}^\top \mathbf{A} \mathbf{S}.$$

This completes the proof. $\square$

# D   Empirical Evidence

We conduct experiments on the NCI1 dataset and plot the heatmap of the coarsened adjacency matrix in Fig. 7, where we can observe that the edge weights in DiffPool may span a wide range due to the involvement of multiple multiplications in their generation. For MinCutPool and DMoNPool, the edge weights are normalized by degree to mitigate numerical explosion. However, this normalization leads to the edge weights becoming excessively smooth and lacking sparsity. Learnable filtration based PH performs effectively on unweighted graphs; however, none of the existing GP methods are capable of appropriately handling the adjacency matrix.

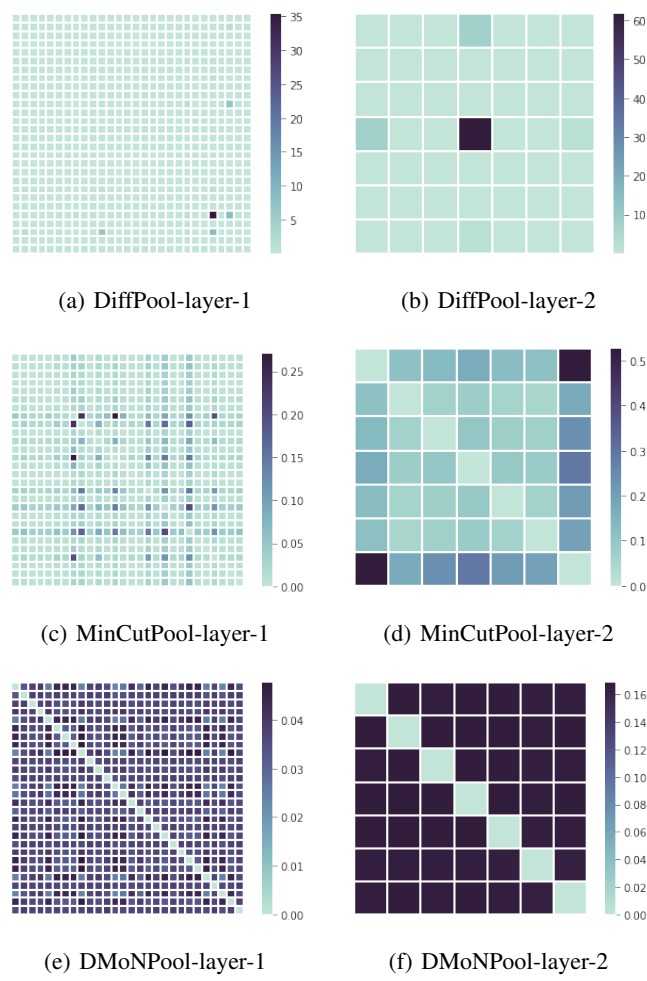

|  |  |
|---|---|
| (a) DiffPool-layer-1 | (b) DiffPool-layer-2 |
| (c) MinCutPool-layer-1 | (d) MinCutPool-layer-2 |
| (e) DMoNPool-layer-1 | (f) DMoNPool-layer-2 |

Figure 7: Heatmap of the coarsened adjacency matrix in terms of DiffPool, MinCutPool, and DMoNPool on NCI1 dataset.

# E   Additional Experiments

## E.1   Visualization of persistence diagrams

We visually represent the 1-dimensional PD of graphs before and after applying TIP in terms of ring and grid2d datasets, as shown in Fig. 8. As described in Appendix B.1, in the original graphs we initialize node features with the eigenvectors of the graph Laplacian matrices. Consequently, the features of different edges exhibit slight variations, resulting in multiple nonoverlapping points in the PDs. Upon applying TIP, we can clearly observe that the one-dimensional topological features related to cycles remain similar to those in the original graphs. This demonstrates TIP's ability to preserve cycles.

## E.2   Running time comparison

We compare the running time (in seconds) of TIP on different datasets. The experiments are conducted using an AMD EPYC 7542 CPU and a single NVIDIA 3090 GPU. We utilize the default settings from the graph classification experiments. We report the average running time of 50 epoches training in Table 6. It is worth noting that TIP is performed $L$ times for $L$ pooling layers, thus the inclusion of TIP does not impose a significant computational burden.

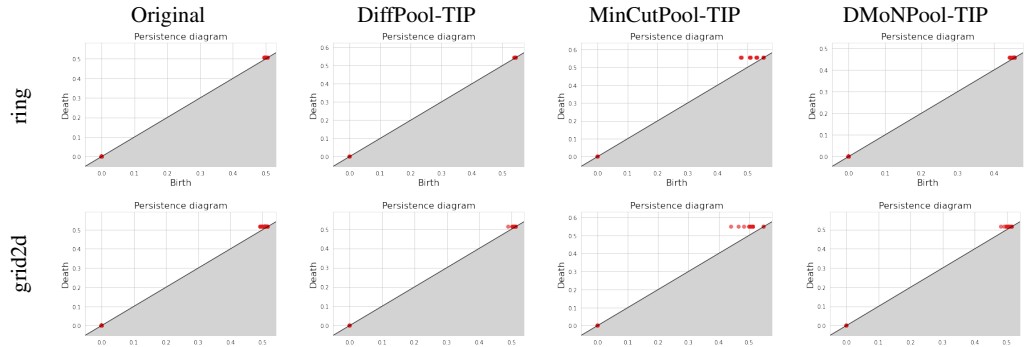

Figure 8: Persistence diagrams of graphs before and after applying TIP in terms of ring and grid2d datasets.

Table 6: Average running time (seconds) comparisons on different datasets.

| Methods | Datasets | | |
|---|---|---|---|
| | NCI1 | PROTEINS | ENZYMES |
| DiffPool | 209.48 | 56.55 | 30.61 |
| DiffPool-TIP | 339.37 | 92.06 | 49.65 |
| MinCutPool | 145.99 | 38.22 | 27.34 |
| MinCutPool-TIP | 296.06 | 79.42 | 41.17 |
| DMoNPool | 124.89 | 35.07 | 19.35 |
| DMoNPool-TIP | 305.63 | 81.34 | 43.82 |

### E.3 Visualization of coarsened graphs without preserving topology

We present some coarsened graphs that do not preserve topology (DiffPool-TIP-NL) in Fig. 9. These graphs contribute equally to the objective in the graph classification task, but their topologies are different. A similar observation was made by [30], who found that randomly generated graphs show equivalent performance. In DiffPool-TIP-NL, other topology-related modules in TIP are preserved, allowing some topological information to be injected into the three results shown in Fig. 9. Guided by the $\mathcal{L}_{topo}$, DiffPool-TIP tends to select the results that are most similar to the original graph among all the options. Experimental results in Fig. 5 demonstrate that this type of topology is superior and leads to better performance on downstream tasks.

### E.4 Topology relevant experiments

To further demonstrate that our proposed method can effectively capture the topological features in graphs, we design an experiment where the topological structure of the graph is highly relevant. We generate a synthetic dataset named Cycles, comprising two balanced 2-class sets of 1000 graphs each. This dataset consists of either a single large cycle (class 0) or two connected large cycles (class 1),

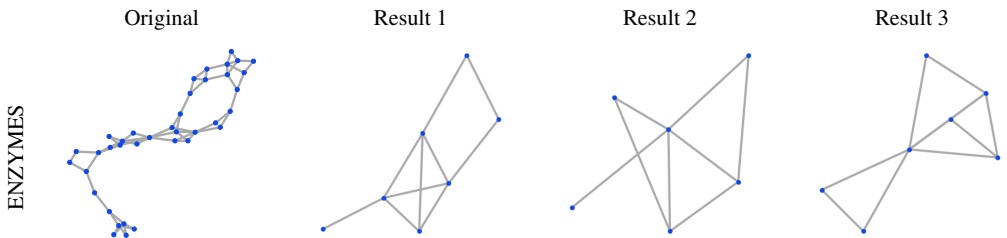

Figure 9: Several coarsened graphs with DiffPool-TIP-NL that contribute equally to the objective.

Table 7: Classification results on synthetic datasets

|  | Cycles | 2-Cycles |
|---|---|---|
| DiffPool | $54.3 \pm 1.1$ | $50.0 \pm 2.2$ |
| DiffPool-TIP | $\mathbf{65.1 \pm 2.7}$ | $51.4 \pm 2.8$ |
| MinCutPool | $54.2 \pm 2.6$ | $49.0 \pm 3.6$ |
| MinCutPool-TIP | $\mathbf{65.0 \pm 2.9}$ | $50.4 \pm 2.9$ |
| DMoNPool | $55.0 \pm 2.7$ | $50.0 \pm 2.6$ |
| DMoNPool-TIP | $\mathbf{68.7 \pm 2.7}$ | $50.0 \pm 3.6$ |

resembling digital numbers "0" and "8", respectively. The distinguishing factor between the classes lies in the presence of cycles, highlighting the significance of the graph's topological structure in classification. The node numbers range from 10 to 20, with 3-dimensional random node features generated. For model configuration, we uniformly use 1-GCN plus 1-pooling layer. The evaluation criteria remain consistent with those outlined in our paper. The experimental results in Table 7 demonstrate the effectiveness of TIP in leveraging topological features to significantly outperform the comparable pooling methods.

Additionally, to evaluate our method's performance on graphs with different number of connected components, we generated a synthetic dataset named 2-Cycles, comprising two balanced two-class sets of 1,000 graphs each. This dataset consists of either two disconnected large cycles (class 0) or two large cycles connected by a single edge (class 1). The distinguishing factor between the classes is the number of connected components. The node numbers range from 10 to 20, with three-dimensional random node features generated. For the model configuration, we uniformly employed one GCN layer plus one pooling layer. Experimental results in Table 7 indicate that our method is not effective in distinguishing similar graphs with different connected components. This aligns with our expectations, as our method does not explicitly incorporate such information, given that most graphs in real-world datasets are connected.

### E.5 Ablation study

To assess the contributions of different modules in our TIP model, we conduct comprehensive ablation studies on NCI1, PROTEINS, ENZYMES, and IMDB-BINARY datasets. We utilize examine five ablated variants of TIP: (i) with no resampling (TIP-NR), (ii) with no persistence injection (TIP-NP), (iii) with no topological loss function (TIP-NL), (iv) with 0-dimensional topological features (TIP-0), (v) with fixed filtration (TIP-F). All these variants are applied on three baseline pooling methods.

As depicted in Table 8, ablating any of the above modules resulte in performance degradation compared to the full model, thus indicating the importance of each designed module in the success of TIP. Additionally, on all three datasets, the resampling module significantly enhance the classification outcomes, while its removal lead to a substantial performance drop. Without resampling, the learnable filtration will treat edges equally, resulting in the inclusion of nonsensical topological information. In some cases, this even impede the model's performance, as observed in the no injection variants which perform worse than their counterparts on the PROTEINS dataset.

Another noteworthy observation is that even in the absence of the topological loss function $\mathcal{L}_{topo}$, GP can still benefit from incorporating PH. This could be attributed to the fact that the learnable filtration can inherently capture certain essential topological information to some extent. Furthermore, our model can still reap the benefits of the topological loss function, which indirectly guides the pooling process, even without explicitly injecting topological information using persistence.

Further, we provide an ablation study of our topological loss term by replacing it with the Wasserstein distance. While the Wasserstein distance is a powerful metric for comparing persistence diagrams, its computation can be computationally intensive, particularly when dealing with high-dimensional vectorized representations. Therefore, it significantly increases our training time in practice. We denote the variant of using Wasserstein distance as "TIP-W". Here we present the ablation study results on two datasets. We can observe that TIP-W has competitive performance compared with the full version TIP (with our proposed loss term), and outperforms the variant TIP-NL (no loss term). Initially, we design our $\mathcal{L}_{topo}$ to avoid the high computational complexity of Wasserstein distance, but

Table 8: Test accuracy of graph classification in ablation study experiments.

| Methods | Datasets | | | |
| --- | --- | --- | --- | --- |
| | NCI1 | PROTEINS | ENZYMES | IMDB-BINARY |
| DiffPool | 77.64 ± 1.86 | 78.81 ± 3.12 | 48.34 ± 5.14 | 73.15 ± 3.30 |
| DiffPool-TIP-NR | 80.82 ± 1.71 | 77.89 ± 4.07 | 55.43 ± 2.81 | 75.00 ± 2.64 |
| DiffPool-TIP-NP | 81.99 ± 1.15 | 79.30 ± 1.26 | 62.22 ± 3.13 | 75.85 ± 2.85 |
| DiffPool-TIP-NL | 82.33 ± 2.14 | 79.11 ± 2.01 | 58.77 ± 5.15 | 76.10 ± 3.78 |
| DiffPool-TIP-W | 83.02 ± 1.08 | 78.25 ± 1.63 | 62.15 ± 4.43 | **76.75 ± 3.66** |
| DiffPool-TIP-0 | 82.45 ± 1.40 | 79.12 ± 1.63 | 56.88 ± 4.96 | 76.25 ± 2.33 |
| DiffPool-TIP-F | 83.21 ± 1.55 | 77.91 ± 3.46 | 60.24 ± 5.15 | 75.75 ± 3.19 |
| DiffPool-TIP | **83.75 ± 1.70** | **79.86 ± 3.12** | **65.05 ± 4.24** | 76.40 ± 3.13 |
| MinCutPool | 77.92 ± 1.67 | 78.25 ± 3.84 | 39.83 ± 2.63 | 73.80 ± 3.54 |
| MinCutPool-TIP-NR | 79.68 ± 1.38 | 78.23 ± 2.92 | 42.51 ± 2.83 | 74.35 ± 1.80 |
| MinCutPool-TIP-NP | 78.81 ± 2.07 | 78.92 ± 3.35 | 45.56 ± 2.81 | 74.65 ± 3.24 |
| MinCutPool-TIP-NL | 78.48 ± 1.86 | 78.40 ± 3.06 | 45.26 ± 4.14 | 74.90 ± 3.03 |
| MinCutPool-TIP-W | 80.06 ± 0.78 | 79.51 ± 4.29 | 46.12 ± 1.23 | 74.50 ± 2.91 |
| MinCutPool-TIP-0 | 78.18 ± 1.34 | 79.64 ± 3.04 | 41.34 ± 1.24 | 74.83 ± 2.41 |
| MinCutPool-TIP-F | 76.65 ± 1.72 | 79.40 ± 3.55 | 44.10 ± 2.68 | 73.80 ± 1.72 |
| MinCutPool-TIP | **80.17 ± 1.29** | **79.73 ± 3.27** | **46.34 ± 3.85** | **75.20 ± 2.67** |
| DMoNPool | 78.03 ± 1.64 | 78.63 ± 3.89 | 40.82 ± 3.68 | 73.50 ± 3.01 |
| DMoNPool-TIP-NR | 79.26 ± 1.01 | 78.72 ± 1.30 | 42.51 ± 4.40 | 73.75 ± 3.30 |
| DMoNPool-TIP-NP | 79.60 ± 0.97 | 79.44 ± 1.68 | 44.36 ± 3.98 | 73.50 ± 3.35 |
| DMoNPool-TIP-NL | 79.08 ± 1.83 | 79.26 ± 1.70 | 43.35 ± 3.90 | 74.00 ± 2.76 |
| DMoNPool-TIP-W | 79.48 ± 1.50 | 79.70 ± 2.95 | 45.45 ± 1.34 | 74.00 ± 2.91 |
| DMoNPool-TIP-0 | 79.23 ± 0.89 | 79.24 ± 3.44 | 41.67 ± 2.04 | 73.60 ± 2.57 |
| DMoNPool-TIP-F | 78.83 ± 1.99 | 79.44 ± 3.39 | 42.88 ± 2.25 | 73.60 ± 2.87 |
| DMoNPool-TIP | **79.68 ± 1.38** | **79.73 ± 3.66** | **45.84 ± 5.32** | **74.25 ± 2.93** |

we are suprised to find that TIP also marginally outperforms TIP-W in numerous instances, potentially attributed to the efficacy of feature transformation and high-order statistical features. These elements serve as a feature augmentation mechanism to enhance the persistence diagrams.

In Section 4.3, we provide theoretical analysis that 1-dimensional topological features are powerful enough to distinguish non-isomorphic graphs, thus eliminating the necessity of incorporating 0-dimensional features. In this section, we provide empirical evidence about incorporating additional 0-dimensional features to support our claim. The results of variant TIP-0 indicates that the inclusion of 0-dimensional topological features merely increases runtime and has no benefits for the overall performance. This explains why we merely consider 1-dimensional topological features in our method.

As for the ablation of filtration functions, we employ an MLP with randomly initialized and fixed parameters as the filtration function. Using learnable filtrations leads to significant gains over random filtration functions in more than half of the cases. In some cases, randomly initialized filtrations may happen to be close to the learned filtrations, but this does not consistently occur.

Overall, our ablation study supports the indispensability and effectiveness of each module in the TIP model, further underscoring their contributions to its success.

### E.6 Evaluation of expressive power

The growing interest in the expressive capability of graph pooling has been prominent in recent studies [3]. A graph pooling model based on GNNs is deemed more effective as it can differentiate a larger set of non-isomorphic graphs by producing unique representations for each. Graph pooling integrated with appropriately designed message-passing layers proves to be as competent as the WL test in distinguishing graphs. Understanding the expressive capacity of graph pooling aids in selecting between existing pooling operators or crafting novel ones. Furthermore, to empirically assess the expressive capacity of our proposed approach, TIP, we conduct experiments on the EXPWL1 dataset

Table 9: Classification results on EXPWL1 dataset.

| Pooling | Test Accuracy |
| --- | --- |
| DiffPool | $97.0 \pm 2.4$ |
| DiffPool-TIP | $\mathbf{99.3 \pm 0.5}$ |
| MinCutPool | $98.8 \pm 0.4$ |
| MinCutPool-TIP | $\mathbf{99.9 \pm 0.1}$ |
| DMoNPool | $99.0 \pm 0.7$ |
| DMoNPool-TIP | $\mathbf{99.7 \pm 0.1}$ |

following the experimental setup detailed in [3]. Each graph pair $(\mathcal{G}_i, \mathcal{H}_i)$ in EXPWL1 consists of two non-isomorphic graphs distinguishable by a WL test, which encode formulas with opposite SAT outcomes. Therefore, any GNN that has an expressive power equal to the WL test can distinguish them and achieve approximately $100\%$ classification accuracy on the dataset. The classification outcomes on the EXPWL1 dataset are shown in Table 9, which reveal the notable improvement in the expressive capacity of graph pooling achieved through our proposed method in empirical evaluations.

