# OpenReview forum: "Boosting Graph Pooling with Persistent Homology"
_NeurIPS.cc/2024/Conference — NeurIPS 2024 poster_

### Official Review · Reviewer_TU8V · 2024-06-26

**Soundness:** 3
**Presentation:** 4
**Contribution:** 4
**Rating:** 8
**Confidence:** 4

**Summary:**

The paper introduces topological graph pooling, a novel method that leverages persistent homology to enhance general graph pooling techniques. The use of persistent homology serves two primary purposes, preserving important topological features in the coarsened graphs and generating edge weights for them. For the first purpose, a topological regularization term comparing the topology of the original and coarsened graphs is developed. For the second purpose, edge weights for the coarsened graph are generated based on an importance score derived from the persistence of cycles containing these edges in the filtration. The authors conducted two sets of experiments to assess the utility of PH in graph pooling. The first set demonstrates that coarsened graphs computed using the proposed PH graph pooling method effectively preserve topological information, even better than other dense and sparse pooling methods. The second set compares and benchmarks the topological pooling enhancement applied to some standard pooling techniques against other pooling techniques and standard graph neural networks. The results, obtained through a comprehensive experimental setup, clearly show the superiority of topological pooling over the other benchmarked methods.

**Strengths:**

This paper is one that evokes a sense of "why didn't I think of that?" due to its natural (but powerful) approach. In my opinion, after addressing a few questions, this paper deserves at least a spotlight at the conference.

**Originality**: As far as I know, the connection between graph pooling and graph filtrations, while conceptually straightforward, has not been explored before.

**Significance**: The persistent homology method addresses a significant problem in graph pooling, as highlighted by the authors. Previously, it was unclear which properties should be preserved in the coarsened graphs after pooling. It was even argued that random pooling could yield performances similar to previous state-of-the-art (SOTA) methods, making unclear why graph pooling was effective and how to design graph pooling techniques appropiately. This work clarifies that preserving the topology of the initial graph is crucial, consistently outperforming other SOTA graph pooling methods across various benchmarks. This makes intuitive sense, as topology concerns global structures, and destroying this information when getting a subgraph seems harmful for the network. This method is not only a clear win for persistent homology in the graph learning field, but also opens new avenues for effective pooling design in GNNs.

**Clarity**: The paper is well-written, with pertinent and informative figures. In particular, Figure 3 effectively illustrates the importance of the contribution.

**Quality**: The experiments are of high quality, encompassing various datasets and configurations. Additionally, the ablation study addressing the effectiveness of topological pooling in preserving input topology is enlightening and supports the use of persistent homology in graph pooling to preserve topology.

**Weaknesses:**

- Figure 1 (a), second row, is strange to me. Usually (and in the pipeline you describe), persistent homology goes from smaller to bigger graphs, and not the inverse path.
- Line 80: Typographical error: "filtraions" should be corrected to "filtrations".
- Equation 1: Missing comma after the formula.
- In lines 114 and 115, you state that filtrations assign each vertex and edge a value. However, this is not always accurate. To have proper simplicial complexes through the filtration, the filtration function must assign lower values to the endpoints of edges than to the values of the edges themselves. Thus, the filtration needs to satisfy $f(\{v_1, v_2\})\geq \max(f(v_1), f(v_2))$.
- In the paragraph starting at line 113, you mention that edges have a filtration value given by $f(\{v_1, v_2\}) = \max(f(v_1), f(v_2))$, but, as stated in the previous comment, you do not restrict edge filtration values to these values. I suggest choosing one approach and modifying this paragraph slightly to ensure consistency.
- In line 123 you state that the persistence diagram $ph(G, f)$ is a set of persistence diagrams $\{D_1, D_2, ...\}$.  However, in other parts, such as Equation 6, you imply that $ph(G, f)$ is only one of these persistence diagrams. I believe this is inconsistent.
- Line 147 uses the word "filtration" twice, which seems redundant. I propose rephrasing it to "The core of PH is the notion of filtration, the selection of which presents a challenging task."

**Important points**: The score I give is subjected to solve/discuss at least the following points:

- Lines 157 and 158 are unclear to me. How do you map the persistence diagrams to edge weights given by births and deaths of persistence diagrams points exactly? I think this should be described formally for several reasons. How do you choose the cycle representatives for the points of persistence diagrams? What does it happen if one edge belongs to more than one cycle? I think this is very relevant, even in the proofs.
- In the proof of Theorem 4.1, I do not understand what are the nodes $u$ and $u'$. What are the nodes with different count? Are they unique? Also, the proof relies on understanding the specific function mapping persistence diagrams to edges. Should the proof end at line 537?
- In the proof of proposition 1, what does it mean to be isomorphic for two graphs with features? I have not seen this definition before.

**Questions:**

- Why do you use the Gumbel-softmax trick? I understand that you obtain a full graph using usual graph pooling, but, precisely, as you are learning filtrations and using persistent homology, you are also capturing in the persistence diagrams the topology of subgraphs that may not need to be fully connected. Did you try to avoid the Gumbel-softmax trick?
- In lines 534, 535, 546, and 541 you write "circles", and not "cycles". Is this intentional? I believe there are more "circles" distributed across the test.
- I have not seen any mention of more general pooling methods, such as cell complex pooling or simplicial complex pooling. Do you think your work can be extended to encompass general pooling on topological deep learning data?
- Have you tried using some fixed filtrations insteaf of learnable filtrations for the graphs benchmark? If so, was there a significant difference compared to learnable filtrations?
- In the complexity section, you mention that persistence diagrams can be computed very efficiently for dimension one. Do you have a reference for this claim? I was aware of this for dimension zero, but I have not encountered it for dimension one before.
- In the Q1 experiments, the filtration is learned during training or is it also fixed as for benchmarking?

**Limitations:**

The limitations addressed are too simple. I think more effort is needed in this aspect.

---

> ### Author Rebuttal · Authors · 2024-08-05
>
> Figure 1 (a), second row, is strange
>
> >Figure 1(a) illustrates that PH and pooling naturally share a similar hierarchical structure, thereby aligning well. It does not imply that PH must progress from smaller to larger graphs.
>
> Typographical errors and incorrect expressions
>
> >We thank Reviewer TU8V for the valuable comments on our notations and expressions, which have significantly enhanced the readability of our paper. We have revised all instances according to the suggestions.
>
> In lines 114 and 115, you state that filtrations assign each vertex and edge a value. However, this is not always accurate.
>
> >We thank Reviewer TU8V for the valuable comments on our statement of filtrations. We apologize for the unclear descriptions about filtrations. In our revised manuscript, we will explicitly state at the beginning of this paragraph that filtration on edges is set as $f(v_1, v_2) = max(f(v_1), f(v_2))$.
>
> Inconsistent descriptions of persistence diagrams in line 123
>
> >We thank reviewer TU8V for the valuable comments. In our revised manuscript, we will add a slice to Eq. (6), e.g. $\mathcal{D}_1 = ph(G, f)[1]$, to keep consistency.
>
> Line 147 uses the word "filtration" twice, which seems redundant
>
> >We thank reviewer TU8V for the valuable comments. In our revised manuscript, we will rephrase this sentence as suggested.
>
> Lines 157 and 158 are unclear
>
> >Edge weights are encoded by persistence (lifespan of each edge: |death_time - birth_time|). Therefore, as stated in Lines 156-158, edges do not form cycles have the same birth time and death time, resulting in zero persistence. Each cycle is paired with the edge that created it, and the other edges in this cycle are assigned a dummy tuple value. If one edge belongs to more than one cycle, only the first time it gives birth to a cycle is considered. These settings are proposed and adopted in [1, 2], and we follow this common practice. Formal descriptions are provided in Appendix B.1, Lines 508-513. To enhance clarity, in the final manuscript, we will move this part to the main paper and modify the relevant sections in detail.
>
> In the proof of Theorem 4.1, I do not understand what are the nodes 𝑢 and 𝑢′. What are the nodes with different count? Are they unique? Also, the proof relies on understanding the specific function mapping persistence diagrams to edges. Should the proof end at line 537?
>
> >Here we mean that we have graphs $\mathcal{G}$ and $\mathcal{G}'$, where nodes $u \in \mathcal{G}$ and $u' \in \mathcal{G}$ are nodes with unique label count in WL test. For example, $\mathcal{G}$ contains 2 green nodes, 1 blue node and 1 red node, then node $u$ stands for the green node. The proof of Theorem 4.1 (PH is at least as expressive as WL) ends at line 537, and the subsequent paragraph demonstrates the existence of some cases where PH is more expressive than WL.  In our revised manuscript, we will add more details for better clarity.
>
> In the proof of proposition 1, what does it mean to be isomorphic for two graphs with features?
>
> >We apologize for our ambiguous description. Here we consider a graph isomorphic test context, where node features denote the initial node labels.
>
> Why do you use the Gumbel-softmax trick?
>
> >There are two reasons. First, the utilized learnable filtration [1] operates on nodes and edges, treating different weights equally. Second, edge weights may span a wide range in usual graph pooling (see Appendix D for empirical evidence). Therefore, we use the Gumbel-softmax trick to resample the subgraph into an unweighted graph. In our ablation study (Appendix E.5), TIP-NR represents our method without the Gumbel-softmax trick, and it demonstrates inferior performance.
>
> Write "circles", and not "cycles".
>
> >We apologize for our inconsistent expressions. Both words refer to the same concept, and we will replace "circles" with "cycles" in our revised manuscript.
>
> I have not seen any mention of more general pooling methods, such as cell complex pooling or simplicial complex pooling. Do you think your work can be extended to encompass general pooling on topological deep learning data?
>
> >Thank you for the insightful comment. We believe that our work has the potential to be extended to encompass general pooling in topological deep learning data, as they also exhibit graph structures and one-dimensional topological features. We leave this extension for future work.
>
> Have you tried using some fixed filtrations insteaf of learnable filtrations
>
> >As suggested by Reviewer fBuZ, we consider using an MLP with randomly initialized and fixed parameters as the filtration function, named as **TIP-F**. The experimental results (see Table S2) on four benchmark datasets show that this kind of filtration is not as good as learnable filtrations in our method.
>
> Reference for complexity analysis
>
> >The complexity analysis about persistent homology has been adequately discussed in [2, 3]. The bottleneck of PH computation is dominated by the complexity of sorting all edges, i.e. $\mathcal{O}(m \log m)$, where $m$ is the numer of edges.
>
> In the Q1 experiments, the filtration is learned during training or is it also fixed as for benchmarking?
>
> >In the Q1 experiments, the filtration is learned during training.  We only keep a fixed filtration in the evaluation of topological similarity.
>
> The limitations addressed are too simple.
>
> >We apologize for our inadequate discussions of the limitations. In addition to the limitations mentioned in our paper, we have found that our method lacks the ability to discriminate between graphs when the number of connected components is the only distinguishing factor. We provide empirical evidence (see Table S4) and analysis in our response to Reviewer 3dRY and will include this in our revised manuscript.
>
> >[1] Graph filtration learning. ICML 2020.
> [2] Topological graph neural networks. ICLR 2022.
> [3] Neural Persistence: A Complexity Measure for Deep Neural Networks Using Algebraic Topology. ICML 2019.

---

> > ### Comment · Reviewer_TU8V · 2024-08-10
> >
> > Thank you very much for addressing all my comments.
> >
> > However, I would need to see a revision of proof 4.1, as I still do not properly understand the proof. In abstract simplicial complexes, self-loops are not allowed (you work with sets, not multisets, so you cannot have an abstract simplicial complex with the multiset {{u, u}}). If you do not consider self-loops, then simply take two graphs containing one and two vertices, respectively. As there are no cycles, there are no points in the 1-persistence diagrams of both graphs, so the graphs are not distinguishable by 1-PH but they are by 1-WL. I think you also need dimension zero here, as in [37] (and then you can use the pairing lemma). Can you point to some references working with simplicial complexes where self-loops are admitted? This is definitely not standard.
> >
> > Regarding the proof of Proposition 1, after carefully rereading it again, I see that you prove that the sum of features is the same, but how can you infer that the two pooled graphs are then the same?
> >
> > These two results are not relevant to the content of the paper itself, but until I either completely understand them or they are removed, I'm afraid I cannot propose an acceptance of the paper. I'm lowering my score to 4 until these two comments are resolved. Sorry for the inconvenience, I thought these two errors were a problem of my own understanding, but now I'm not so sure.

---

> > > ### Author Response · Authors · 2024-08-10
> > > **Thanks to Reviewer TU8V and some extra clarification**
> > >
> > > Q: In abstract simplicial complexes, self-loops are not allowed.
> > > > Thank you for pointing out our inaccurate description. We acknowledge that standard simplicial complex does not consider self-loops. In our method, we are not trying to modify the computation process of PH to make it adapt to graphs with self-loops. We simply augment the 1-dimensional persistence diagrams by puting the self-loops on the diagonal of persistence diagrams obtained by **standard PH**. Therefore, the principles of simplicial complex and PH are not destroied.
> > > >
> > > >We apologize for our unclear descriptions about this part. In our revised manuscript, we will term the **self-loop augmented 1-dimensional persistence diagram** as $\mathcal{\tilde{D}}_1$, and make clear descriptions such that readers can be aware that $\mathcal{\tilde{D}}_1 = \mathcal{D}_1 + \mathrm{self loops}$ rather than  $\mathcal{\tilde{D}}_1 = ph(G+\mathrm{self loops})$.
> > >
> > > Q: I would need to see a revision of proof 4.1
> > > > We provide a **proof sketch** of Theorem 4.1. We first assume the existence of a sequence of WL labels and show how to construct a filtration function $f$ from this. Consider nodes $u$ and $u'$ are nodes with unique count in $\mathcal{G}$ and $\mathcal{G'}$, then our filtration is constructed such that their filtration values $f(u)$ and $f(u')$ are unique and different. Consider all three cases: (1)$u$ and $u'$ are both in cycles; (2) $u$ and $u'$ are both not in cycles; (3) one of $u$ and $u'$ is in cycles and the other is not. For all the cases, $f(u)$ and $f(u')$ will be revealed in their respective persistence diagrams. Since $f(u)$ and $f(u')$ are unique and different, we can use the persistence diagrams to distinguish the two graphs.
> > > >
> > > > We apologize for our unclear proof of the theorem. Note that in the three cases considered above, **self-loops and cycles are still considered separately**. Therefore, **our proof of Theorem 4.1 is based on the self-loop augmented 1-dimensional persistence diagram, which is still within the scope of abstract simplicial complex**. In our revised manuscript, we will first give a proof sketch, and then reorganize our proof into several steps for better clarity.
> > >
> > > Q: In Proposition 1, the sum of features is the same, but how can you infer that the two pooled graphs are then the same?
> > > > As we consider labelled graphs (with node features), Proposition 1 is about the isomorphic invariant property over such graphs, where we assume that two graphs are isomorphic, i.e. for two graphs $G_1$ and $G_2$ with nodes features $X$ and $Y$, there exists a permutation $P$such that $X = P(Y)$ . It's clear that $\sum{X_i} = \sum{Y_i}$ at this point. The proof of Proposition 1 (Eq. (9)) states that given equal sum of node features (basic settings of isomorphic graphs), the sum of node features are still equal after pooling. **As stated in Lines 551-556, all the operations are permutation equivariant, so after pooling the graph connectivities are still isomorphic to each other.** Therefore, the pooled graphs are still isomorphic to each other.
> > > >
> > > >We apologize for our unclear descriptions that cause ambiguity. In our revised manuscript, we will add an additional paragraph to emphasize the permutation equivariant property of all operations in pooling, specifically:
> > > (Informal proof) Let $(G:=(N,E),A)$ be a graph with connectivity $A$. Consider pooling $A' = B A B^\top$, where $B^\top=\mathrm{GNN}(G)$ is an assignment. If we permute $G$ using a permutation matrix $P$, then $B^\top P^\top = \mathrm{GNN}(GP)$ (because GNNs are permutation-equivalent for input node-level tasks) and the permutated connectivity is $PAP^\top$. Thus the permuted graph after pooling is $A'=B^\top P^\top P A P^\top PB$, which means that isomorphic graphs after pooling are still isomorphic.
> > > >
> > > >We hope that this can help readers understand that after pooling the graph connectivities are still isomorphic to each other.
> > >
> > > > Thank you for your feedback and for bringing these concerns to our attention. We apologize for any confusion these results may have caused. We have carefully reviewed the issues you've highlighted and will make the necessary revisions to either clarify or remove the results in question. Your insights are important to us, and we will work diligently to address them in our revised submission.

---

> > > > ### Comment · Reviewer_TU8V · 2024-08-11
> > > >
> > > > Now we agree on Theorem 4.1. Thank you very much for your answer! Please, modify also the statement of 4.1 to avoid saying that PH is enough (as you are proposing with the new term).
> > > >
> > > > Last question: Could you be more concrete on how you use the equality of the sums for the isomorphism? From the argument you are giving to me right now, I do not clearly see where you use Equation (9) in the isomorphism proof.
> > > >
> > > > Again, thank you for your answer, raising the score to 8 again!

---

> ### Author Response · Authors · 2024-08-12
> **Thanks to Reviewer TU8V and some extra clarification**
>
> Thank you for your understanding and for raising the score. We appreciate your careful consideration of Theorem 4.1 and will revise the statement to ensure it accurately reflects the necessary conditions.
>
> Regarding your final question, we apologize for any lack of clarity in our previous explanation. Initially, we consider a graph level task (e.g. graph classification). In this case, we consider if the sum of node features are invariant for any permutation of the graph, then it is isomorphism invariant (refer to Equation (9)). Now we provide more general proof of the invariant property.
>
> **Proof.** Following the notations in our last response, we prove isomorphic invariance at feature level and *connectivity* level.
> For **feature-level invariance**, let $X \in \mathbb{R}^{n \times d}$ be the node features, $P \in \\{0,1\\}^{n \times n}$ be the permutation matrix, $B\in \mathbb{R}^{n' \times n}$ be the assignment matrix, and $P^{\top}X$ be the permutated node features. The node feature map after pooling is denoted as $X' \in \mathbb{R}^{n' \times d}$, then we have
> $$X' = BX$$
> If we permute $G$ using a permutation matrix $P$, the permutated node features after pooling are $$X' = (BP)(P^{\top}X)$$
> which proves the isomorphism invariant property of pooling at feature level.
> For **connectivity-level invariance**, we copy our previous responses here.
> $$A'=(B P) (P^\top A P) (P^\top B^\top)$$
> Note in our previous response with *informal proof*, we made a mistake in $A'=B^\top P^\top P A P^\top PB$ due to a rush, where $B^\top P^\top$ does not has a compatible dimension. Besides, for symmetric connectivity $A$, $PAP^\top=P^\top AP$. We correct it in this version.
> This completes the proof.
> We will revise our paper up to this new proof accordingly. Thank you once again for your valuable feedback and support.

---

### Official Review · Reviewer_HRv2 · 2024-07-12

**Soundness:** 2
**Presentation:** 2
**Contribution:** 2
**Rating:** 6
**Confidence:** 2

**Summary:**

This paper proposes a TDA-based mechanism for storing phase information in the pooling layer of a GNN. The proposed approach resamples the connections in the graph and scales the edge weights using persistence information from a one-dimensional persistence diagram. Experiments on artificial data for concept evaluation and on several benchmark datasets were also carried out.

**Strengths:**

- The retention of phase information in the pooling layer is a new and interesting approach.
- Experiments have shown that performance can be improved.

**Weaknesses:**

- The proposed method uses a 1-dim betti number, but its validity is not known. All 1-dim death times seem to be the same (maximum time for filtration) and seem to be only information about the last connection of the cycle. Only limited information on the cycle seems to be stored and is less convincing.
- Learning to preserve topology information in persistent diagrams is proposed in [1]. (It would be inappropriate not to mention this.)  This paper vectorises PDs for training, but [1] and [2] show the possibility of learning in a more data-loss-free form, which lacks consideration. The novelty of the method also seems to be limited to the introduction of phase conservation in the pooling layer.
- The paper only considers TU Datasets, ZINC and OGB, which have become mainstream in recent evaluations, should also be included in this evaluation.

[1] Topological Autoencoders, ICML 2020

[2] Optimizing persistent homology based functions, ICML 2021

**Questions:**

See Weakness

**Limitations:**

The authors adequately addressed the limitations.

---

> ### Author Rebuttal · Authors · 2024-08-05
>
> W1: The proposed method uses a 1-dim betti number, but its validity is not known. All 1-dim death times seem to be the same (maximum time for filtration) and seem to be only information about the last connection of the cycle. Only limited information on the cycle seems to be stored and is less convincing.
>
> >It's common practice to use PH to characterize 0 and 1 dimentional betti numbers and intergrate with GNNs [1, 2, 3, 4], which has been proven effective. Following the practical implementations in previous works mentioned above, the 1-dim death times are equal and set as a large constant value, but their birth times vary according to their filtration values, resulting in different persistences (|death_time - birth_time|). We adhere to the conventions in [1] and utilize persistence rather than extended persistence for efficiency.
>
> W2: Learning to preserve topology information in persistent diagrams is proposed in [5]. (It would be inappropriate not to mention this.) This paper vectorises PDs for training, but [5] and [6] show the possibility of learning in a more data-loss-free form, which lacks consideration. The novelty of the method also seems to be limited to the introduction of phase conservation in the pooling layer.
>
> >We thank the reviewer for providing another perspective and evidence to support our claim that preserving topology information is meaningful in pooling. Our motivation arises from the observation that PH and graph pooling naturally aligns well, as evidenced in Figure 1. Preserving topology information in general can also be considered as graph pooling, as noisy topology is filtered out and essential parts are preserved. This is the major innovation of our method. We apologize for the unclear statement of our motivation and will reorganize and include new references in our revised manuscript.
> >
> >We acknowledge that [5] and [6] provide effective ways to preserve topology information. However, directly calculating the distance between PDs is also a stable, expressive, and practical way to assess the similarity of two graphs, as proved by [4]. Additionally, vectorizing PDs is a common practice [1, 3, 7], which we adopt for its efficiency and flexibility. Autoencoders emphasize upstream tasks and therefore have higher demands for data loss, while pooling focuses on downstream tasks and has no such requirement.
>
> W3: The paper only considers TU Datasets, ZINC and OGB, which have become mainstream in recent evaluations, should also be included in this evaluation.
>
> >We have already considered one OGB dataset MOLHIV, as shown in the last column of Table 2. To address your concern, we provide additional experimental results on the ZINC dataset (see Table S1) in our response to Reviewer fBuZ.
>
> >[1] Topological graph neural networks. ICLR 2022.
> [2] Graph filtration learning. ICML 2020.
> [3] Deep learning with topological signatures. NIPS 2017.
> [4] Curvature filtrations for graph generative model evaluation. NIPS 2023.
> [5] Topological autoencoders. ICML 2020.
> [6] Optimizing persistent homology based functions. ICML 2021.
> [7] Persistence enhanced graph neural network. AISTATS 2020.

---

> > ### Comment · Reviewer_HRv2 · 2024-08-09
> >
> > Thank you for the clarification.
> > Regarding the first point: the authors argue that this is not a problem because the method is generally known to be good, but this does not seem to be an appropriate response, as even if it is generally good, it does not necessarily mean that it is good with regard to the subject in question.
> > I assume that only insufficient information on the timing of cycle generation for 1-dim remains. However, after reading the comments, it occurred to me that they might be using an extended diagram that combines 0-dim and 1-dim. If so, it is enough to check that the text does not cause a misunderstanding, because perhaps a misunderstanding has just occurred.
> > On the second point: this question concerns novelty. Assuming we have recognised that each is appropriate, we would like to clarify which parts are major novel contributions. The main novelty is the concept of the topology preservation of the proposal, and each tool is existing, but is the contribution of the specific construction of an effective combination among them, or does each tool also have its own challenges and is there novelty in those areas as well? This is important for determining the level of novelty and at present we recognise it as the former. In your opinion, what are your views on this?

---

> > > ### Author Response · Authors · 2024-08-09
> > > **Thanks to Reviewer HRv2 and some extra clarification**
> > >
> > > Thank you for your insightful feedback. We are happy to address your concerns and questions. Detailed responses to your comments are provided below.
> > >
> > > Q1:
> > > > We apologize for any previous confusion. Initially, we have an observation in Figure 1(a) that PH and graph pooling both seek to coarsen a given graph in a hierarchical fashion, which motivates us to conduct **a preliminary experiment**, which uses a pioneer graph pooling method to perform graph classification while simultaneously computing the persistence of the coarsened graph. Interestingly, **we observed a monotonic trend between the pooling ratio and non-zero 1-dimensional persistence is commonly shared by a wide range of graph data**, as evidenced in Figure 1(b). We emphasize that we are the first to report these meaningful findings. This underscores the validity of using PH in graph pooling due to their shared structural patterns, thus motivating us to integrate topological features into graph pooling. Moreover, extensive experiments (see Table 2) also demonstrate that 1-dimensional topological features are valid in boosting three classic dense pooling methods.
> > > >
> > > >We acknowledge that we extended persistence diagrams to increase expressivity, by storing node-related information in self-loops. We apologize for any unclear descriptions. To avoid ambiguity, we will make the following major modifications to our revised manuscript:
> > > >1. In Section 4.2, we will add preliminary experiments in Table S3 (in the attached PDF in our General Response) to explain why we do not directly incorporate ordinary 0-dimensional features. Subsequently, the self-loop augmented 1-dimensional persistence diagram is denoted as $\tilde{\mathcal{D}}_1$.
> > > 2. In Section 4.3, we will revise our claim to state "self-loop augmented 1-dimensional topological features computed by PH are sufficient enough to be more expressive than 1-WL."
> > >
> > >
> > > Q2:
> > > >Thank you for raising this question. Our novelty and contribution encompass the following 3 aspects, extending beyond merely combining existing tools effectively:
> > > >1. **Findings**. We are the first to have the findings that PH and graph pooling naturally aligns well across multiple datasets (see Figure 1(b)), which motivates us to use PH to boost graph pooling, and we believe, may be advisable for related methodical designs.
> > > 2. **Methodology**. Guided by our findings, we developed three specific modules in our methodology:
> > > >    - Given that coarsened graphs in dense graph pooling are always fully connected  (see Figures 3, 4) with widely ranging edge weights  (see Appendix D for empirical evidence),  we designed a differentiable **Resampling** process to make the coarsened graphs well suited for using learnable filtrations. This paves the way for the following designed modules in effectively injecting topological information into graph pooling.
> > > >    - Utilizing the resampled graphs from last step, we designed a **Persistence Injection** module, capitalizing on the alignment between persistence and graph pooling.
> > > >    - We designed a **Topological Loss Function**. This can be viewed as a neural surrogate to regularize that topological structures should be preserved in graph pooling.
> > >   Experimental results in Appendix E.5 demonstrate that without our Resampling process, directly combining learnable filtrations with graph pooling may hinder model performance in some cases. Persistence Injection and Topological Loss also proved effective for pooling. Hence, from a methodological standpoint, we contribute a straightforward yet effective way to integrate PH with graph pooling.
> > > 3. **Theoretical Analysis**. We demonstrate that self-loop augmented **1-dimensional** persistence diagrams can boost the expressive power of graph pooling methods.
> > > We acknowledge that our previous clarification was somewhat weak. In our revised manuscript, we will modify our summary of contributions in the Introduction and reorganize Section 4.2 for improved comprehension of our contribution.
> > > >
> > > >[1] Hofer, Christoph, et al. "Graph filtration learning." International Conference on Machine Learning. PMLR, 2020.

---

> > > > ### Comment · Reviewer_HRv2 · 2024-08-10
> > > >
> > > > I understood that the algorithm was built on the basis of the ' finding' from various validations that the 1-dim birthpoints, i.e. the timing of cycle generation, correlate with the present aim. It is very interesting that the TDA is highly effective despite dropping a lot of information from the TDA. If you ask me, it makes sense that information on cycles is important for pooling, but I hope that it will be clarified why it is so effective as an issue for the future. If the architecture is based on that finding and is designed to that effect, then that is a novelty (if it is clarified). I would like to raise my score because it clarified the content and gave very interesting findings.

---

> > > > > ### Author Response · Authors · 2024-08-10
> > > > > **Thanks to Reviewer HRv2**
> > > > >
> > > > > Thank you very much for your insightful and encouraging feedback! We are delighted that our response has addressed your concerns about the novelty. Your comments about our interesting findings are highly appreciated.
> > > > >
> > > > > We fully agree that further clarification on the effectiveness of information about cycles is essential. Based on your valuable suggestions, we will revise our paper to better elucidate this aspect and further highlight the contributions and novelty of our approach.
> > > > >
> > > > > Your comments have significantly helped us to better present the core innovations of our work. We sincerely appreciate your recognition and thoughtful consideration in raising the score.

---

### Official Review · Reviewer_3dRY · 2024-07-12

**Soundness:** 2
**Presentation:** 3
**Contribution:** 2
**Rating:** 6
**Confidence:** 4

**Summary:**

This paper proposes a topology-based graph pooling layer, TIP. TIP fits easily with the current graph pooling frameworks. Once the pooled graph is obtained from any existing graph pooling techniques, the authors make this pooled graph adaptable to persistent homology. They consider the PH of this graph and optimize it by minimizing the topological loss function between the current persistence diagram and  original persistence diagram. The authors show experimental results on some synthetic and real-world datasets.

**Strengths:**

1. It is, indeed, a new idea to combine PH for graph pooling to maintain topological information in the graph during pooling.

2. Experiment to show that the proposed method preserves important topological features is well-chosen.

3. The paper provides experimental evidence that addition of TIP into a standard GNN framework improves the performance on most datasets for graph classification.

**Weaknesses:**

1. Theorem 4.1 is incorrect as stated. Only the 1-dimensional topological features computed by PH cannot be as expressive as 1-WL test in distinguishing non-isomorphic graphs. Consider two graphs with no loops, with different numbers of vertices. 1-WL test will be able to distinguish these graphs while 1-dimensional topological features computed by PH will not be able to. One would need to use the 0-dimensional topological features computed by PH. Refer to [1]. Adding self-loops is just a proxy for counting the number of vertices, which can be done using 0-dimensional features.

2. Moreover, the authors, in [1], have already shown that PH is at least as expressive as 1-WL test. Hence, the theoretical contribution of the paper does not seem significant.

[1]: Topological Graph Neural Networks, Horn et.al

**Questions:**

1. What happens when you consider graphs with more than one component? How does TIP perform in that scenario compared to other pooling mechanisms?

2. Have you tried other ways to incorporate $L_{topo}$? For e.g., by choosing a different vectorization of PDs such as persistence images or rational hats?

**Limitations:**

Yes, the authors have discussed limitations.

---

> ### Author Rebuttal · Authors · 2024-08-05
>
> W1: Theorem 4.1 is incorrect as stated. Only the 1-dimensional topological features computed by PH cannot be as expressive as 1-WL test in distinguishing non-isomorphic graphs. Consider two graphs with no loops, with different numbers of vertices. 1-WL test will be able to distinguish these graphs while 1-dimensional topological features computed by PH will not be able to. One would need to use the 0-dimensional topological features computed by PH. Refer to [1]. Adding self-loops is just a proxy for counting the number of vertices, which can be done using 0-dimensional features.
>
> >As stated in our proof of Theorem 4.1, adding self-loops results in additional cycles, which are reflected in the diagonal of one-dimensional persistence diagrams (PDs). Consequently, the augmented one-dimensional PDs can distinguish graphs with different numbers of nodes, as they correspond to different numbers of points in the one-dimensional PDs. Therefore, this distinction can be achieved using 0-dimensional or 1-dimensional features. Our method can be easily extended to integrate both 0-dimensional and 1-dimensional features, which we denote as TIP-0. Preliminary experiments, shown in Table S3, indicate that the inclusion of zero-dimensional topological features merely increases runtime. Thus, our proposed method focuses solely on 1-dimensional PDs.
>
> W2: Moreover, the authors, in [1], have already shown that PH is at least as expressive as 1-WL test. Hence, the theoretical contribution of the paper does not seem significant.
>
> >In our response to reviewer fBuZ, we have stated that our theoretical novelty lies in the further proof that PH with **1-dimensional** features is also as expressive as WL, while [1] only proved **0-dimensional** case. Moreover, we theoretically proved the **isomorphic invariant property** of our method.
>
> Q1: What happens when you consider graphs with more than one component? How does TIP perform in that scenario compared to other pooling mechanisms?
>
> >To address your concern, we generated a synthetic dataset named 2-Cycles, comprising two balanced two-class sets of 1,000 graphs each. This dataset consists of either two disconnected large cycles (class 0) or two large cycles connected by a single edge (class 1). The distinguishing factor between the classes is the number of connected components. The node numbers range from 10 to 20, with three-dimensional random node features generated. For the model configuration, we uniformly employed one GCN layer plus one pooling layer. The evaluation criteria remained consistent with those outlined in our paper. Experimental results in Table S4 indicate that our method is not effective in distinguishing similar graphs with different connected components. This aligns with our expectations, as our method does not explicitly incorporate such information, given that most graphs in real-world datasets are connected.
>
> Q2: Have you tried other ways to incorporate 𝐿𝑡𝑜𝑝𝑜? For e.g., by choosing a different vectorization of PDs such as persistence images or rational hats?
>
> >Thanks for the valuable comment. There are many topological descriptors, and we simply follow previous works [1, 2] and tried some of them. To make the best of PDs, we use several transformations and concatenate the output of them, including triangle point transformation, Gaussian point transformation and line point transformation. We have not tried rational hats and are grateful for reviewer 3dRY's suggestion. However, concerning the time budget of rebuttal, we would leave this as future work.
>
> >[1] Graph filtration learning. ICML 2020.
> [2] Topological graph neural networks. ICLR 2022.

---

> > ### Comment · Reviewer_3dRY · 2024-08-10
> >
> > I thank the authors for their efforts.
> >
> > However, I am not convinced by Theorem 4.1. Adding self-loops seems forced, just to fix the case of different number of nodes.
> >
> > Moreover, in the proof of Proposition 1, graphs can be isomorphic with the node features being different, right? The filtration function is S_n-equivariant. But how does that guarantee that the summation of node features on two different graphs is equal to, to begin with?
> >
> > I appreciate the efforts put into the experiment about different number of connected components.
> >
> > Hence, as of now, I would like to stick to my score.

---

> > > ### Author Response · Authors · 2024-08-10
> > > **Thanks to Reviewer 3dRY and some extra clarification**
> > >
> > > Q: However, I am not convinced by Theorem 4.1. Adding self-loops seems forced, just to fix the case of different number of nodes.
> > > > In our proof of Theorem 4.1, we categorize the cases into 3 categories: (1)$u$ and $u'$ are both in cycles; (2) $u$ and $u'$ are both not in cycles; (3) one of $u$ and $u'$ is in cycles and the other is not. Adding self-loops as augmentation is used to handle case (2). Theoretically, **this simple yet effective augmentation eliminates the necessity of computing 0-dimensional topological features, thus reducing computational burdens**. Furthermore, empirical results (Table S3 in the attached PDF in our General Response) showed that **using 0-dim topological features merely increases runtime**, so our proposed method focuses only on 1-dim persistence diagrams.
> > >
> > > Q: In the proof of Proposition 1, graphs can be isomorphic with the node features being different, right?
> > > > We apologize for any ambiguity caused by our unclear descriptions. In the context of graph isomorphism, node features refer to node labels (e.g. node colors or degrees). Therefore, for isomorphic graphs with node features (labels) $X$ and $Y$, there exists a permutation $P$ such that $X = P(Y)$. Under this setting, the sum of node features is equal. If graphs have totally different node features, they cannot be considered as isomorphic graphs.
> > > In our revised manuscript, we will **replace "node features" with "node labels"**, and provide detailed descriptions of the graph isomorphim settings.
> > >
> > > >Thank you for your feedback and for pointing out these concerns. We apologize for any confusion the results may have caused. After carefully reviewing the issues you highlighted, we will make the necessary revisions. Your insights are valuable to us, and we are committed to addressing them thoroughly in our revised submission.

---

> > > ### Author Response · Authors · 2024-08-12
> > > **Thanks to Reviewer 3dRY and some extra clarification**
> > >
> > > Thank you for your continued interest in our paper. We noticed that you have concerns regarding the proof of Proposition 1. We apologize for the lack of clarity in our previous proof. In our latest response to Reviewer TU8V, we have provided a more general and formal proof, which you might find interesting.

---

> > > > ### Comment · Reviewer_3dRY · 2024-08-12
> > > >
> > > > Dear Authors,
> > > >
> > > > Thank you for your response. Yes, I read your response to reviewer TU8V, and now it makes much more sense. As far as Theorem 4.1 goes, I still have my reservations. However, I understand that this result is novel because now it uses augmented 1-dimensional persistence information only. I do not have any further questions. I have revised my score.

---

> > > > > ### Author Response · Authors · 2024-08-12
> > > > > **Thanks to Reviewer 3dRY**
> > > > >
> > > > > Thank you for taking the time to review our response and for your thoughtful feedback throughout this process. We are glad to hear that our explanation clarified our approach
> > > > >
> > > > > We appreciate your understanding regarding the novelty of our theorem. Your insights have been invaluable in helping us refine our work, and we are grateful for your willingness to engage with our response and adjust your score accordingly.
> > > > >
> > > > > Thank you again for your constructive feedback and for your efforts in reviewing our submission.

---

### Official Review · Reviewer_k5BH · 2024-07-12

**Soundness:** 3
**Presentation:** 2
**Contribution:** 3
**Rating:** 7
**Confidence:** 5

**Summary:**

This paper proposes a new and systematic way to integrate persistent homology into GNNs for improved performance.  Rather than applying PH in a brute force manner as much existing work does, this work adapts the method based on the coincidence between the graph pooling mechanism and the filtration of PH.  The work takes advantage of the dynamic nature of PH via the filtration and results in improved performance in message passing.

**Strengths:**

This was a very nice paper that studies an important problem and proposes a novel approach using persistent homology.  In my mind, this is perhaps among the first most convincing applications and demonstrations of the relevance of persistent homology to deep learning theory and implementations.

**Weaknesses:**

The writing style of the paper could be improved, benefitting from a spell and linguistics check.

**Questions:**

GNNs are known to encompass other NN models and architectures, such as CNNs and transformers.  Is the approach adaptable to these special cases of GNNs?  Does the performance also hold up in machine learning tasks, such as image classification?  Would the theoretical guarantees also need to be adapted?

**Limitations:**

Limitations were considered in the check list, however, I would like to see these concerns integrated into the paper.  In addition, it would be helpful to have examples and demonstrations of the limitations of the method.

---

> ### Author Rebuttal · Authors · 2024-08-05
>
> Questions: Is the approach adaptable to these special cases of GNNs? Does the performance also hold up in machine learning tasks, such as image classification? Would the theoretical guarantees also need to be adapted?
>
> >The proposed approach is adaptable as long as the data can be represented as graphs with specific topological structures. There is no need to modify the theoretical guarantees.
>
> Limitations: I would like to see these concerns integrated into the paper. In addition, it would be helpful to have examples and demonstrations of the limitations of the method.
>
> >Thanks for your valueable comment. In our revised manuscript, we will relocate the discussion of limitations to Section 6, Conclusion. The limitations are evident: tree-like graphs have no cycles, rendering the one-dimensional persistence diagram meaningless. Additionally, in our response to Reviewer 3dRY, we provide supplementary experiments on synthetic datasets, demonstrating that one limitation of our method is its inability to distinguish between graphs when the number of connected components is the only differentiating factor.

---

> > ### Comment · Reviewer_k5BH · 2024-08-10
> > **Acknowledgment of rebuttal**
> >
> > I have read reports by other reviewers and the authors' responses, as well as seen the additional experiments and proposed improvements.  I maintain my opinion that this a strong paper that I would like to see accepted in the conference and therefore am maintaining my positive rating.

---

> > > ### Author Response · Authors · 2024-08-10
> > > **Thanks to Reviewer k5BH**
> > >
> > > Thank you for taking the time to carefully review our additional experiments and responses. We sincerely appreciate your continued positive assessment and your support for our work. Your thoughtful feedback has been invaluable in helping us refine our paper, and we are grateful for your recommendation.

---

### Official Review · Reviewer_fBuZ · 2024-07-15

**Soundness:** 2
**Presentation:** 2
**Contribution:** 2
**Rating:** 4
**Confidence:** 4

**Summary:**

The paper introduces TIP (topology-invariant pooling), a PH-based pooling layer. The proposed approach involves resampling graph connections from soft-cluster assignment matrices and adjusting edge weights using persistence information derived from 1-dimensional diagrams. TIP leverages a loss function designed to maintain topology by leveraging vectorizations of the 1-dimensional diagrams. Experimental evaluations conducted on synthetic and real classification datasets illustrate the performance of the proposed method.

**Strengths:**

- Flexibility: TIP is agnostic to the GNN method and can be easily combined with GNN layers
- Novelty: This combination of PH for graph pooling tasks is novel.

**Weaknesses:**

- The theoretical part of the paper does not seem very relevant and novel. Theorem 4.1 only states the **existence** of a filtration that is at least as expressive as the 1-WL test, which has limited impact in practice. Similar results are well-known in the literature (e.g., [1]). Also, in the analysis, we must allow filtration functions f to leverage graph structure, not only node features as defined in the paper. Finally, it seems that Theorem 4.1 can be easily turned into a strictly more expressive statement (see question below).
- Another main concern revolves around the motivation for preserving cycles in general-purpose graph coarsening methods. Overall, I found the discussion/motivation for this design choice hand-wavy.
- While the proposed method improves over the base pooling layers (e.g., DiffPool), it does not lead to significant gains over some (pooling-free) models in Table 2. Also, the experiments include only 1 OGB dataset --- I recommend adding ZINC or at least another OGB dataset.

**Questions:**

1. For the non-zero persistence tuples, are the death times always equal $\infty$ (or f_max + constant)? If so, isn't TIP exploiting only (non-persistent) homology?
2. Figure 2 does not show self-loops after the resampling procedure.
3. Including the definition of simplicial complexes (SCs) would be helpful. Are graphs (with self-loops) 1-dimensional SCs? Do self-loops count as independent cycles?
4. How can the hierarchical view of PH in Figure 1 be obtained from sublevel filtrations as defined in Section 3? In other words, how can we achieve a decreasing sequence of subgraphs using sublevel filtrations?
5. Where does the proposed method's expressivity stand compared to other graph pooling layers (see [2])? The Appendix provides a brief discussion. I think that discussion could be added to the main paper.
6. Consider the same setting as Theorem 4. Isn't PH based on $D_1$ **strictly** more expressive than 1-WL? To prove this additional gain, it suffices to show a pair of graphs $G, G'$ with $D_1(G) \neq D_1(G')$ that 1-WL cannot distinguish. Wouldn't G=two triangles and G'=hexagon (where all nodes have the same features) be an example of such a pair?
7. How does the proposed model perform using a fixed (non-trainable) random filtration function (e.g., an MLP where the parameters are fixed)?

[1] - Topological GNNs, ICLR 2022.

[2] - The expressive power of pooling in Graph Neural Networks, NeurIPS 2023.

**Limitations:**

The paper mentions as limitation the "heavy reliance of the proposed method on circular structures within graphs" in Appendix F.

---

> ### Author Rebuttal · Authors · 2024-08-05
>
> W1:Concerns about theory and filtrations.
>
> >The theoretical analysis of the expressive power and other properties of graph pooling is crucial in this field, as it aids in selecting between existing pooling operators or developing new ones [1]. Beyond TOGL [2] that proves the expressivity of PH using **0-dim** features, our theoretical novelty lies in the further proof that PH with **1-dim** features is also as expressive as WL.  Additionally, in Appendix E.6, empirical results demonstrate that our proposed method can be more effective in distinguishing non-isomorphic graphs, indicating that our model can learn to approximate these filtrations in practice.
> >
> >Furthermore, we apply the filtration function f to the hidden representations $\mathbf{X}^{(l)}$ obtained through GNNs (see Eq. (4) and (6)), where graph structures are considered. This type of operation is also found in relevant literature [2, 5].
>
> W2: Motivation for preserving cycles
>
> >In graph pooling, the notion of a good/useful latent graph topology is less obvious. Many efforts have been devoted to learning representations with additional regularizers, among which PH is frequently used for data with topological structures. Since preliminary experiments (in Table S3) showed that using 0-dim topological features merely increases runtime, our proposed method focuses only on 1-dim persistence diagrams, which is explained as preserving cycles in graph coarsening. This claim is also confirmed by Reviewer HRv2 with literature evidence [3, 4].
>
> W3: No significant gains over some pooling-free models. Experiments on ZINC.
>
> >Our primary objective is to devise a novel mechanism to **boost** graph pooling methods rather than proposing a new method to surpass existing GNN models. Our performance depends on how the original pooling methods perform. Experimental results demonstrate that the proposed method achieves substantial performance improvement when applied to several pooling methods. Additionally, our method outperforms strong pooling-free baselines such as GSN in all but one case.
> >
> >Furthermore, we conducted an additional experiment on ZINC dataset. Following the settings in [7], we used the graph-level prediction task, and the results of MAE are shown in Table S1. Our models consistently outperform three dense pooling methods, demonstrating their ability to combine the benefits of both graph pooling and persistent homology.
>
> Q1: For the non-zero persistence tuples, are death times always equal
> >It is a well-established practice to extend persistence diagrams to mitigate numerical issues, as demonstrated in previous studies [2, 6]. The persistence tuples with non-zero values have identical death times but distinct birth times determined by their filtration values, resulting in varying persistences. Consequently, TIP leverages persistent homology to enhance graph pooling methods.
>
> Q2: Fig. 2 does not show self-loops
> >We have revised Fig. 2. The new version is provided in PDF.
>
> Q3: Definition of simplicial complexes.
> >A brief definition of simplicial complexes has already been discussed in Sec 3, lines 107-112. Graphs with self-loops can be considered low-dimensional simplicial complexes containing only 0-simplices (vertices) and 1-simplices (edges). Self-loops are counted as cycles.
>
> Q4: The hierarchical view of PH in Figure 1
> >In persistent homology, given a filtration function $f$, we can get a finite set of values $a_n > \cdots > a_1$ and generate a sequence of nested subgraphs of the form $\mathcal{G} = \mathcal{G}_n  \supseteq \ldots  \mathcal{G}_k \ldots \supseteq \mathcal{G}_0 \supseteq \emptyset$, where $\mathcal{G}_k = (V_k, E_k)$ is a subgraph of $\mathcal{G}$ with $V_k:=$ { $v \in V \mid f(\mathbf{x}_v) \leq a_k$} and $E_k:=${$(v, w) \in E \mid \max (f(x_v), f(x_w)) \leq a_k$}. This process is interpreted as the hierarchical view of PH in Fig. 1. As the value decreases, a decreasing sequence of subgraphs is obtained.
>
> Q5: Expressivity issues
> >A theoretical discussion is provided in Sec 4.3, lines 209-213, asserting that the proposed method is more expressive than dense pooling methods, which are discussed in this paper. Additional empirical evaluation of expressive power is presented in Appendix E.6. In our revised manuscript, the empirical conclusions will be integrated into the main paper in Section 4.3.
>
> Q6: Isn't PH based on 𝐷1 strictly more expressive than 1-WL?
> >Our final conclusion aligns with your statement. The proof consists of two stages: First, we prove that the expressive power of PH is at least as strong as WL, which is Theorem 4.1; Second, we provide specific examples, as you mentioned, where PH exhibits greater expressive power than WL. Thus, the final conclusion is that PH's expressive power surpasses that of WL. In the discussion following Theorem 4.1 and in Appendix C, we present statements similar to yours. We apologize for any ambiguity in our previous statements. In our revised manuscript, we will integrate the two stages into Theorem 4.1.
>
> Q7: What if using a fixed (non-trainable) random filtration function?
> >Thank you for the valuable comment, which is instrumental in demonstrating the effectiveness of using learnable filtrations in our method. Following your suggestions, we employ an MLP with randomly initialized and fixed parameters as the filtration function, named as **TIP-F**. Additional experiments are conducted on four benchmark datasets, as shown in Table S2. Overall, learnable filtrations outperform TIP-F, although in some cases, TIP-F achieves similar performance.
>
> >[1] The expressive power of pooling in graph neural networks. NIPS 2023.
> [2] Topological graph neural networks. ICLR 2022.
> [3] Topological autoencoders. ICML 2020.
> [4] Connectivity-optimized representation learning via persistent homology. ICML 2019.
> [5] Graph filtration learning. ICML 2020.
> [6] Deep learning with topological signatures. NIPS 2017.
> [7] Rethinking pooling in graph neural networks. NIPS 2020.

---

> > ### Comment · Reviewer_fBuZ · 2024-08-13
> >
> > I thank the authors for their efforts to address my concerns, especially for the additional experiments.
> >
> > However, most of my concerns remain. More details here:
> >
> > > Beyond TOGL [2] that proves the expressivity of PH using 0-dim features, our theoretical novelty lies in the further proof that PH with 1-dim features is also as expressive as WL.
> >
> > PH with 1-dim features is as expressive as 1-WL because the incorporation of self-loops corresponds to adding the birth time of the 0-dim diagrams to the 1-dim diagrams --- information from the birth times (1-WL colors) suffices to be as expressive as 1-WL. Also, note that the setting in TOGL differs from the self-looped case in this paper. Therefore, I find it misleading to say that the theoretical results of this paper go beyond what is in  [2]. I think the theoretical results are rather trivial.
> >
> > > we apply the filtration function f to the hidden representations $\mathbf{X}^{(l)}$ obtained through GNNs (see Eq. (4) and (6)), where graph structures are considered.
> >
> > The domain of the filtration function is not consistently used throughout the paper. In line 114, f is a function on nodes + edges. In line 120, f is a function on the initial features of the vertices. In the proofs, f leverages the 1-WL colors. Thus, to be exact, the filtration f should take the graph structure as input, not only the initial features (as in line 120). This is what I meant by my comment.
> >
> > > The persistence tuples with non-zero values have identical death times but distinct birth times determined by their filtration values, resulting in varying persistences. Consequently, TIP leverages persistent homology to enhance graph pooling methods.
> >
> > I disagree. If all non-zero persistent tuples have death times equal to infinity, then they are all essential features that are captured by homology (Betti 1) + the values of the filtering functions. To capture the persistence of independent cycles, people have applied Extended PH [1].
> >
> > [1] PersLay: A Neural Network Layer for Persistence Diagrams and New Graph Topological Signatures, AISTATS 2020.
> >
> > > A brief definition of simplicial complexes has already been discussed in Sec 3, lines 107-112
> >
> > I don't think lines 107-112 provide a proper definition.
> >
> > > we can get a finite set of values $a_n > \cdots > a_1$ and generate a sequence of nested subgraphs of the form $\mathcal{G} = \mathcal{G}_n \supseteq \ldots \mathcal{G}_k \ldots \supseteq \mathcal{G}_0 \supseteq \emptyset$, where $\mathcal{G}_k = (V_k, E_k)$ is a subgraph of $\mathcal{G}$ with $V_k:=$ { $v \in V \mid f(\mathbf{x}_v) \leq a_k$} and $E_k:=${$(v, w) \in E \mid \max (f(x_v), f(x_w)) \leq a_k$}. As the value decreases, a decreasing sequence of subgraphs is obtained.
> >
> > "As the value decreases" --- what does this mean when we have that $a_1 < a_2 < ... < a_k ... < a_n$? Isn't $k$ the index for the sequence? Also, in section 3, the paper considers the pre-image of [-\infty, a]. I still believe there is a clear mismatch between the motivation in Figure 1 and the idea of sub-level filtrations used in the paper.
> >
> > Finally, the performance gains over random filtering functions are mostly marginal.
> >
> > Therefore, I would like to keep my initial rating.

---

> > > ### Author Response · Authors · 2024-08-13
> > > **Thanks to Reviewer fBuZ and some extra clarification**
> > >
> > > Thank you for your insightful feedback. We are happy to address your concerns and questions. Detailed responses to your comments are provided below.
> > >
> > > > Q1
> > >
> > > We apologize for any unclear clarification. We acknowledge that it's inappropriate to state that our theoretical result goes beyond [1]. Rather, we claim that it is an extension of the theoretical results in TOGL with extra **practical** meanings. The novelty of our theoretical result lies in that by augmenting the 1-dimensional persistence diagrams with self-loops, **the necessity of explicitly computing 0-dimensional persistence diagrams is eliminated, thus reducing computational burdens**. This strongly supports our algorithm design and the novelty is also confirmed by Reviewer 3dRY. We will clarify this point in the revised version according to your suggestion.
> > >
> > > > Q2
> > >
> > > We apologize for any unclear descriptions. We acknowledge that we have a slightly abuse of notations. In line 114, we want to express that generally filtration functions can be either node-based or edge-based. As the adopted filtrations are node based, so in the rest of this paper we mainly use the node-based filtration functions.
> > > In line 120 and the rest of the paper, $f(x_v)$ does not imply that filtration functions are used on the initial node features, but to express that it is a node-based filtraiton. In our previous response, we explained that the filtration function is applied on the hidden repressentations obtained through GNNs. To avoid ambiguity, in our revised manuscript, we will add another variable $h_v$ as the hidden representation and replace $f(x_v)$ with $f(h_v)$ to distinguish it from the initial node feature $x_v$.
> > >
> > > > Q3
> > >
> > > We employed ordinary PH mainly because it has been proven to be effective in capturing cycle-related topological information [1, 2], and main focus of this paper is to design a framework to boost graph pooling with PH rather than extend existing filtration learning methods. As claimed in [1],  it is also possible to use extended persistence, but extended PH may cause additional computational burden in our setting. Considering all these factors, extended PH was not adopted in our paper. Indeed, as pointed by the reviewer, incorporating extended PH may hopefully bring about additional benefit, thus we would investigate this in our future work.
> > >
> > > > Q4
> > >
> > > We apologize for our previous clarification. The definition of simplicial complex is essential in the context of TDA, and we provide formal definitions here, which will be added to our revised paper:
> > > **Definition 1 (Simplicial Complex)**: A simplicial complex $K$ consists of a set of simplices of certain dimensions. Each simplex $\sigma \in K$ has a set of faces, and each face $\tau \in \sigma$ has to satisfy $\tau \in K$. An element $\sigma \in K$ with $∣\sigma∣=k+1$ is called a $k$-simplex, which we denote by writing $\mathrm{dim} \sigma = k$. Furthermore, if $k$ is maximal among all simplices in $K$, then $K$ is referred to as a $k$-dimensional simplicial complex. A graph can be seen as a low-dimensional simplicial complex that only contains 0-simplices (vertices) and 1-simplices (edges).
> > >
> > > > Q5
> > >
> > > We apologize for our previous clarification that causes misunderstanding. The ordered filtration values $a_1 < \cdots < a_n$ in Section 3 is not directly related to Figure 1(a). In Figure 1(a), we are actually considering **persistence** (lifespan of each tuple, see Figure 1(c)) in PH rather than the PH process itself. As high persistence corresponds to features and low persistence is typically considered noise, when we gradually filter out edges with low persistence, a hierarchical view of subgraphs is obtained. This process shares a similar hierarchical fashion with graph pooling, where the filtering of low persistence can be viewed as dropping unimportant edges in graph pooling, as illustrated in Figure 1(a). In our methodology, the _Persistence Injection_ module implements the low persistence filtering operation, in line with our motivation.
> > > In our revised manuscript, we will modify Figure 1(a) by replacing PH with **persistence filtering** to prevent ambigurity. Moreover, we will provide illustrative examples for a better understanding of the persistence filtering process.
> > >
> > > > The performance gains over random filtering functions are mostly marginal.
> > >
> > > Using learnable filtrations leads to significant gains over random filtration functions in more than half of the cases. In some cases, randomly initialized filtrations may happen to be close to the learned filtrations, but this does not consistently occur. Similar findings have been made in graph pooling that **randomly pooled subgraphs** also leads to good performance compared to **learnable subgraphs** [3]. We thank Reviewer fBuZ for proposing the implementation of random filtrations that promotes the new finding, as this area remains unexplored. This is out of the scope of this paper, and we plan to explore this finding in our future work.

---

> > > ### Author Response · Authors · 2024-08-13
> > > **Thanks to Reviewer fBuZ and some extra clarification**
> > >
> > > Thank you for your feedback and for bringing these concerns to our attention. We apologize for any confusion our previous clarification may have caused. We have carefully reviewed the issues you've highlighted and will make the necessary revisions. Your insights are important to us, and we will work diligently to address them in our revised submission.
> > >
> > > [1] Topological graph neural networks. ICLR 2022.
> > > [2] Graph filtration learning. ICML 2020.
> > > [3] Rethinking pooling in graph neural networks. NIPS 2020.

---

### Author Rebuttal · Authors · 2024-08-05

**General Response:**

We would like to express our sincere gratitude for your thorough review of our manuscript and for providing valuable feedback and suggestions. Your expertise and insights have been instrumental in improving the quality and clarity of our work.

During the rebuttal period, __we provide additional experimental results (Tables) and revised figure in a PDF file__. All experimental codes will be updated and released accordingly. Specifically:

1. **Theory.** Reviewer fBuZ, 3dRY, and TU8V raised concerns about our theorem. We would like to emphasize that the innovation of our theory lies in further proving that the expressive power of one-dimensional topological features is stronger than that of the Weisfeiler-Lehman (WL) test. To clarify our proof of theorem, we will reorganize and explain it step by step.
2. **Motivations.** Reviewer fBuZ is concerned about our motivations for preserving cycles in pooling. Meanwhile, Reviewer HRv2 provides literature evidence supporting the preservation of topological features. Our motivation arises from the observation that persistent homology (PH) and graph pooling naturally align well. This motivates us to preserve topological features in the pooling process. Preliminary experiments (shown in Table S3) indicated that using zero-dimensional topological features merely increases runtime; hence, our method focuses on one-dimensional persistence diagrams, which is explained as preserving cycles.
3. **Experiments.** Both Reviewers fBuZ and HRv2 suggested adding experiments on the ZINC dataset. We have provided results in Table S1, where a consistent improvement is observed when integrating our method. To address the concerns of Reviewer fBuZ and TU8V about using fixed filtration functions, we use an MLP with randomly initialized and fixed parameters as the filtration function, as suggested by Reviewer fBuZ. Experimental results in Table S2 demonstrate that it is more effective to use learnable filtrations rather than fixed ones.
4. **Limitations.** To address Reviewer 3dRY's question about our method's performance on graphs with more than one component, we generated a synthetic dataset named 2-Cycles. As most graphs contain only one connected component and we do not explicitly incorporate such information, our method does not show significant improvements for multi-component graphs, as evidenced in Table S4. This can be considered a limitation of our method, which we also discussed in response to Reviewer TU8V.

We sincerely appreciate the time and effort you have devoted to reviewing our manuscript and providing constructive feedback. Your contributions have significantly strengthened our research.

## **PDF**

---

### Author Response · Authors · 2024-08-14

While some discussions did not come to a close, we sincerely thank all reviewers for their efforts in providing constructive comments and valuable feedback, which have significantly improved our manuscript!

---

### Decision · Program_Chairs · 2024-09-25

**Decision:**

Accept (poster)

**Comment:**

This paper presents a novel scheme for pooling graphs, making use of recently-developed topological paradigms, i.e., persistent homology. Using multi-scale representations of graphs (based on so-called filtration functions), the paper develops a new method for topology-aware graph coarsening, which is shown to lead to improved outcomes in an experimental suite. The paper is a timely and relevant contribution to a growing body of work on methods that leverage topological concepts in machine learning. Given the described benefits, the submission thus has the potential to be of high impact, not only for graph learning, but also as a general signal for the utility of topology-based methods.

This assessment was reflected in the reviews and the ensuing discussion. There was near-unanimous agreement on the relevance and soundness of the proposed work, but some reviewers raised concerns about theoretical and empirical aspects. As part of an extensive rebuttal, it is the impression of the AC—and of the reviewers—that these concerns could be alleviated. There is thus agreement that the paper is ready for presentation at the conference.

In their revised version, the authors should focus on the following aspects:

- Discuss delineation to existing methods better in the main paper (reviewer ` fBuZ`)
- Better discussion of limitations (reviewer ` k5BH`)
- Discussing additional data sets (reviewer `HRv2`), in particular those that have been mentioned as part of the extensive rebuttal)

The authors are moreover encouraged to heed the advice concerning the overall presentation of the work, with multiple reviewers pointing out suggestions for improvement. I trust the authors to provide these changes and turn this submission into an impactful contribution at the conference.